# Hybrid working from home improves retention without damaging performance

Nicholas Bloom[1,5] ✉, Ruobing Han[2,5] ✉ & James Liang[3,4] ✉

Working from home has become standard for employees with a university degree. The most common scheme, which has been adopted by around 100 million employees in Europe and North America, is a hybrid schedule, in which individuals spend a mix of days at home and at work each week[1,2]. However, the effects of hybrid working on employees and firms have been debated, and some executives argue that it damages productivity, innovation and career development[3–5]. Here we ran a six-month randomized control trial investigating the effects of hybrid working from home on 1,612 employees in a Chinese technology company in 2021–2022. We found that hybrid working improved job satisfaction and reduced quit rates by one-third. The reduction in quit rates was significant for non-managers, female employees and those with long commutes. Null equivalence tests showed that hybrid working did not affect performance grades over the next two years of reviews. We found no evidence for a difference in promotions over the next two years overall, or for any major employee subgroup. Finally, null equivalence tests showed that hybrid working had no effect on the lines of code written by computer-engineer employees. We also found that the 395 managers in the experiment revised their surveyed views about the effect of hybrid working on productivity, from a perceived negative effect (−2.6% on average) before the experiment to a perceived positive one (+1.0%) after the experiment. These results indicate that a hybrid schedule with two days a week working from home does not damage performance.

Working from home (WFH) surged after the COVID-19 pandemic, with university-graduate employees typically WFH for one to two days a week during 2023 (refs. 2,6). Previous causal research on WFH has focused on employees who are fully remote, usually working on independent tasks in call-centre, data-entry and helpdesk roles. This literature has found that the effects of fully remote working on productivity are often negative, which has resulted in calls to curtail WFH[5–12]. However, there are two challenges when it comes to interpreting this literature. First, more than 70% of employees WFH globally are on a hybrid schedule. This group comprises more than 100 million individuals, with the most common working pattern being three days a week in the office and two days a week at home[2,8,9]. Second, most employees who are regularly WFH are university graduates in creative team jobs that are important in science, law, finance, information technology (IT) and other industries, rather than performing repetitive data-entry or call processing tasks[10,11].

This paper addresses the gap in previous studies in two key ways. First, it uses a randomized control trial to examine the causal effect of a hybrid schedule in which employees are allowed to WFH two days per week. Second, it focuses on university-graduate employees in software engineering, marketing, accounting and finance, whose activities are mainly creative team tasks.

Our study describes a randomized control trial from August 2021 to January 2022, which involved 1,612 graduate employees in the Airfare and IT divisions of a large Chinese travel technology multinational called Trip.com. Employees were randomized by even or odd birthdays into the option to WFH on Wednesday and Friday and come into the office on the other three days, or to come into the office on all five days.

We found that in the hybrid WFH ('treatment') group, attrition rates dropped by one-third (mean$_{control}$ = 7.20, mean$_{treat}$ = 4.80, $t(1610)$ = 2.02, $P$ = 0.043) and work satisfaction scores improved (mean$_{control}$ = 7.84, mean$_{treat}$ = 8.19, $t(1343)$ = 4.17, $P$ < 0.001). Employees reported that WFH saved on commuting time and costs and afforded them the flexibility to attend to occasional personal tasks during the day (and catch up in the evenings or weekends). These effects on reduced attrition were significant for non-managerial employees (mean$_{control}$ = 8.59, mean$_{treat}$ = 5.33, $t(1215)$ = 2.23, $P$ = 0.026), female employees (mean$_{control}$ = 9.19, mean$_{treat}$ = 4.18, $t(568)$ = 2.40, $P$ = 0.017) and those with long (above-median) commutes (mean$_{control}$ = 6.00, mean$_{treat}$ = 2.89, $t(609)$ = 1.87, $P$ = 0.062).

At the same time, we found no evidence of a significant effect on employees' performance reviews, on the basis of null equivalence tests, and no evidence of a difference in promotion rates over periods of up to two years ('Null results' section of the Methods). We did find significant differences in pre-experiment beliefs about the effects of WFH on productivity between non-managers and managers. Before

[1]Department of Economics, Stanford University, Stanford, CA, USA. [2]Shenzhen Finance Institute, School of Management and Economics, The Chinese University of Hong Kong, Shenzhen, China. [3]National School of Development, Peking University, Beijing, China. [4]Trip.com, Shanghai, China. [5]These authors contributed equally: Nicholas Bloom, Ruobing Han. ✉e-mail: nbloom@stanford.edu; hanruobing@cuhk.edu.cn; liangjz@trip.com

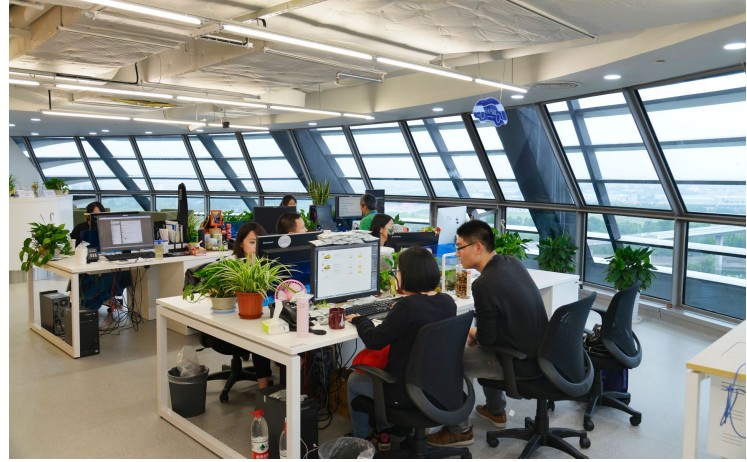
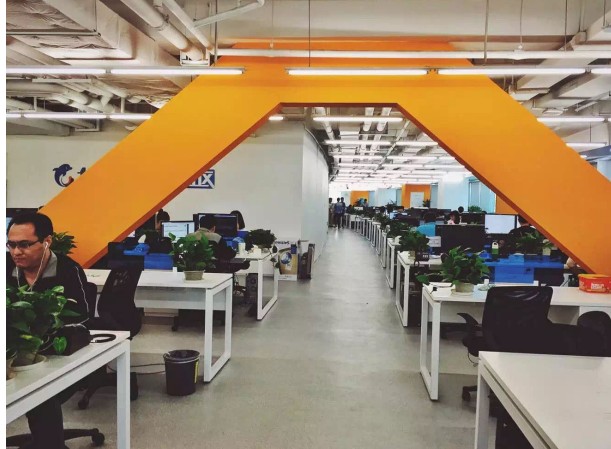

**Fig. 1 | Trip.com employees worked in modern open-plan offices, with teams seated together.** Pictures of Trip.com employees in the office during the experiment. The people in the experimental sample are typically in their mid-30s, and 65% are male. All of them have a university undergraduate degree and 32% have a postgraduate degree, usually in computer science, accounting or finance, at the master's or PhD level. They have 6.4 years tenure on average and 48% of employees have children (Extended Data Table 1).

the experiment, managers tended to have more negative views, reporting that hybrid WFH would be likely to affect productivity by −2.6%, whereas non-managers had more positive views (+0.7%) ($t(1313) = -4.56$, $P < 0.001$). After the experiment, the views of managers increased to +1.0%, converging towards non-managers' views ($\text{mean}_{\text{non-manager}} = 1.62$, $\text{mean}_{\text{manager}} = 1.05$, $t(1343) = -0.945$, $P = 0.345$). This highlights how the experience of hybrid working leads to a more positive assessment of its effect on productivity—consistent with the overall experience in Asia, the Americas and Europe throughout the pandemic, where perceptions of WFH improved considerably[13].

## The experiment

The experiment took place at Trip.com, the third-largest global travel agent by sales in 2019. Trip.com was established in 1999, was quoted on NASDAQ in 2003 and was worth about US$20 billion at the time of the experiment. It is headquartered in Shanghai, with offices across China and internationally, and has roughly 35,000 employees.

In the summer of 2021, Trip.com decided to evaluate the effects of hybrid WFH on the 1,612 engineering, marketing and finance employees in the Airfare and IT divisions, spanning 395 managers and 1,217 non-managers. All experimental participants were surveyed at baseline, with questions on expectations, background and their interest in volunteering for early participation in the experiment. The firm randomized employees with an odd-number birthday (born on the first, third, fifth and so on day of the month) into the treatment group.

Figure 1 shows two pictures of employees working in the office to highlight three points. First, in the second half of 2021, COVID incidence rates in Shanghai were so low that employees were neither masked nor socially distanced at the office. Although the COVID pandemic had led to lockdowns in early 2020 and during 2022, during the second half of 2021, Shanghai employees were free to come to work, and typically were unmasked in the office. Second, employees worked in modern open-plan offices in desk groupings of four or six colleagues from the same team, reflecting the importance of collaboration. Third, the office is a large modern building, similar to many large Asian, European and North American offices.

## Effects on employee retention

One key motivation for Trip.com in running the experiment was to evaluate how hybrid WFH affected employee attrition and job satisfaction. The net effect was to reduce attrition over the experiment by 2.4%, which against the control-group base of 7.2% was a one-third (33%) reduction in attrition ($\text{mean}_{\text{control}} = 7.20$, $\text{mean}_{\text{treat}} = 4.80$, $t(1610) = 2.02$, $P = 0.043$). Consistent with this reduction in quit rates, employees in the treatment group also registered more positive responses to job-satisfaction surveys ($\text{mean}_{\text{control}} = 7.84$, $\text{mean}_{\text{treat}} = 8.19$, $t(1343) = 4.17$, $P < 0.001$). Employees were anonymously surveyed on 21 January 2022, and employees in the treatment group showed significantly higher scores on a scale from 0 (lowest) to 10 (highest) in 'work–life balance', 'work satisfaction', 'life satisfaction' and 'recommendation to friends', and significantly lower scores in 'intention to quit' (Extended Data Table 2).

One possible explanation for the lower quit rates in the treatment group is that quit rates in the control group increased because the individuals in this group were annoyed about being randomized out of the experiment. However, quit rates in the same Airfare and IT divisions were 9.8% in the six months before the experiment—higher than the rate for the control group during the experimental period. Quit rates over the experimental period in the two other Trip.com divisions for which we have data (Business Trips and Marketing) were 10.5% and 9.8%—again higher than that for the control group during the experimental period. This suggests that, if anything, the control-group quit rates were reduced rather than increased by the experiment, possibly because some of them guessed (correctly) that the policy would be rolled out to all employees once the experiment ended.

Figure 2 shows the change in attrition rates by three splits of the data. First, we examined the effect on attrition for the 1,217 non-managers and 395 managers separately. We saw a significant drop in attrition of 3.3 percentage points for the non-managers, which against a control-group base of 8.6% is a 40% reduction ($\text{mean}_{\text{control}} = 8.59$, $\text{mean}_{\text{treat}} = 5.33$, $t(1215) = 2.23$, $P = 0.026$). By contrast, there was an insignificant increase in attrition for managers ($\text{mean}_{\text{control}} = 2.96$, $\text{mean}_{\text{treat}} = 3.13$, $t(393) = -0.098$, $P = 0.922$). We also found that non-managers were more enthusiastic before the experiment, with a volunteering rate of 35% (versus 22% for managers), matching the media sentiment that although non-managerial employees are enthusiastic about WFH, many managers are not ($t(1610) = 4.86$, $P < 0.001$).

Second, we examined the effect on attrition by total commute length, splitting the sample into people with shorter and longer total commutes on the basis of the median commute duration (two-way commutes of 1.5 h or less versus those exceeding 1.5 h, with 648 and 611 employees, respectively). We found that there was a larger reduction in quit rates (52%) for those with a long commute ($\text{mean}_{\text{control}} = 6.00$, $\text{mean}_{\text{treat}} = 2.89$, $t(609) = 1.87$, $P = 0.062$). The reduction in quit

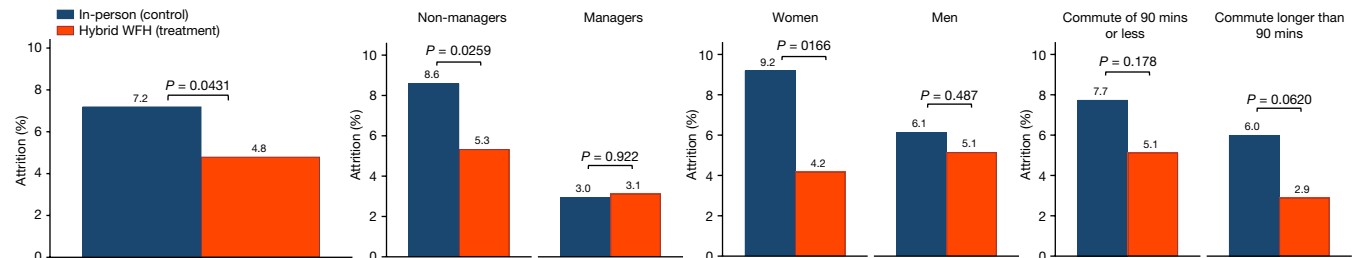

**Fig. 2 | WFH cut attrition by 33% overall, and had a particularly strong effect for non-managers, women and those with longer commutes.** Data on 1,612 employees' attrition until 23 January 2022. Top left, all employees. Only 1,259 employees filled out the baseline survey question on commuting length, so the commute-length (two ways) sample is for 1,259 employees. Sample sizes are 820 and 792 for control and treatment; 1,217 and 395 for non-managers and managers; 570 and 1,042 for women and men; and 648 and 611 for short and long commuters, respectively. Two-tailed $t$-tests for the attrition difference within each group between the control and treatment groups are

(difference = 2.40, s.e. = 1.18, confidence interval (CI) = [0.0748, 4.72], $P$ = 0.043) for all employees; (difference = 3.26, s.e. = 1.46, CI = [0.392, 6.12], $P$ = 0.026) for non-managers; (difference = −0.169, s.e. = 1.73, CI = [−3.57, 3.23], $P$ = 0.922) for managers; (difference = 5.01, s.e. = 2.08, CI = [0.915, 9.10], $P$ = 0.017) for women; (difference = 0.997, s.e. = 1.43, CI = [−1.82, 3.81], $P$ = 0.487) for men; (difference = 2.61, s.e. = 1.93, CI = [−1.19, 6.41], $P$ = 0.178) for employees with median (90 min, two-way) or shorter commutes; and (difference = 3.11, s.e. = 1.66, CI = [−0.156, 6.37], $P$ = 0.062) for above-median (90 min, two-way) commuters.

rates was similarly large for employees with a long commute if we instead defined a long commute as a two-way commute time exceeding 2 h (mean$_{control}$ = 7.33, mean$_{treat}$ = 1.89, $t$(307) = 2.31, $P$ = 0.021). Employees who volunteered to take part in the experiment had longer one-way commute durations (Extended Data Table 3; mean$_{non-volunteer}$ = 0.80, mean$_{volunteer}$ = 0.89, $t$(1257) = −3.68, $P$ < 0.001). This is not surprising given that the most frequently cited benefit of WFH is no commute[1].

Third, we examined the effect on attrition by gender, examining the 570 female and 1,042 male employees separately. We found that there was a 54% reduction in quit rates for female employees (mean$_{control}$ = 9.2, mean$_{treat}$ = 4.2, $t$(568) = 2.40, $P$ = 0.017). For male employees, there was an insignificant 16% reduction in quit rates (mean$_{control}$ = 6.15, mean$_{treat}$ = 5.15, $t$(1040) = 0.70, $P$ = 0.487). This greater reduction in quit rates among female individuals echoes the findings of previous studies[6,14–16], which suggest that women place greater value on remote work than men do. Notably, although the treatment effect of WFH was significantly larger for female employees, volunteers were less likely to be female (mean$_{non-volunteer}$ = 0.37, mean$_{volunteer}$ = 0.32, $t$(1610) = −2.02, $P$ = 0.043); this might suggest that women have greater concerns about negative career signalling by volunteering to WFH.

## Employee performance and promotions

Another key question for Trip.com was the effect of hybrid WFH on employee performance. To assess that, we examined four measures

of performance: six-monthly performance reviews and promotion outcomes for up to two years after the start of the experiment, detailed performance evaluations, and the lines of code written by the computer engineers. We also collected self-assessed productivity effects of hybrid working from experimental participants before and after the experiment to evaluate employee perceptions.

Performance reviews are important within Trip.com as they determine employees' pay and career progression, so are carefully conducted. The review process for each employee is built on formal assessments provided by their managers, co-workers, direct reports and, if appropriate, customers. They are reviewed by employees, collated by managers and by the human resources team, and then discussed between the manager and the employee. This lengthy process takes several weeks, providing a well-grounded measure of employee performance. Although these reviews are not perfect, given their tight link to pay and career development, both managers and employees put a large amount of effort into making these informative measures of performance.

Figure 3 reports the distribution of performance grades for treatment and control employees for the four half-year periods: July to December 2021, January to June 2022, July to December 2022 and January to June 2023. These four performance reviews span a two-year period from the start of the experimental period. Across all review periods, we found no difference in reviews between the treatment and control groups (Extended Data Table 4 and 'Null results' section of the Methods).

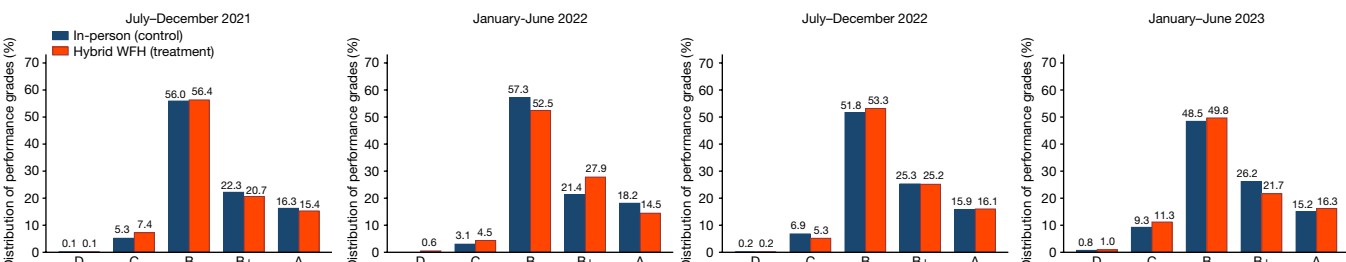

**Fig. 3 | WFH had no significant effect on performance reviews over the next two years.** Results from performance reviews of 1,507 employees in July–December 2021, 1,355 employees in January–June 2022, 1,301 employees in July–December 2022 and 1,254 employees in January–June 2023. Samples are lower over time owing to employee attrition from the original experimental sample. Two-tailed $t$-tests for the performance difference within each period between the control and treatment groups, after assigning each letter

grade a numeric value from 1 (D) to 5 (A), are (difference = 0.056, s.e. = 0.043, CI = [−0.029, 0.14], $P$ = 0.198) for July–December 2021; (difference = 0.034, s.e. = 0.044, CI = [−0.0529, 0.122], $P$ = 0.440) for January–June 2022; (difference = −0.019, s.e. = 0.046, CI = [−0.11, 0.072], $P$ = 0.677) for July to December 2022; and (difference = 0.046, s.e. = 0.051, CI = [−0.054, 0.146], $P$ = 0.369) for January–June 2023. The null equivalence tests are included in the 'Null results' section of the Methods.

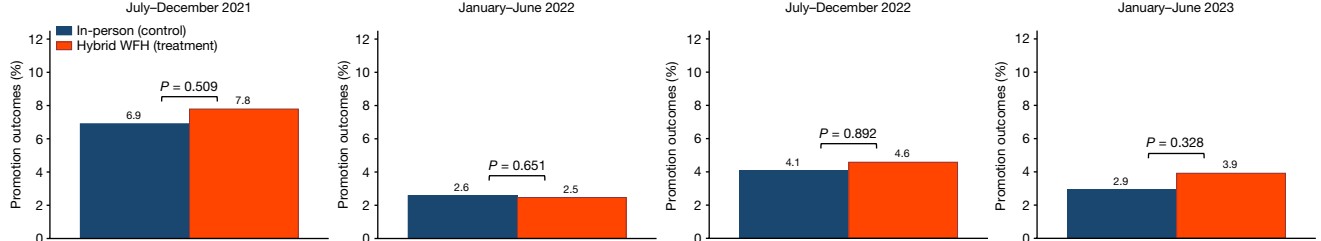

**Fig. 4 | WFH had no significant effect on promotions over the next two years.** Promotion outcomes for 1,522 employees in July–December 2021, 1,378 employees in January–June 2022, 1,314 employees in July–December 2022 and 1,283 employees in January–June 2023. Samples are lower over time owing to employee attrition from the original experimental sample. Two-tailed *t*-tests for the promotion difference within each period between the control and treatment groups are (difference = −0.86, s.e. = 1.34, CI = [−3.51, 1.74],

*P* = 0.509) for July–December 2021 promotions; (difference = 0.12, s.e. = 0.85, CI = [−1.54, 1.78], *P* = 0.892) for January–June 2022 promotions; (difference = −0.51, s.e. = 1.12, CI = [−2.72, 1.70], *P* = 0.651) for July–December 2022 promotions; and (difference = −0.99, s.e. = 1.02, CI = [−2.99, 1.00], *P* = 0.328) for January–June 2023 promotions. The null equivalence tests are included in the 'Null results' section of the Methods.

Figure 4 reports the distribution of promotion outcomes for the treatment and control employees for the same periods. We see no evidence of a difference in promotion rates across treatment and control employees. This is an important result given the evidence that fully remote working can damage employee development and promotions[14,17,18].

We also analysed the effects of treatment on performance grades and promotions for a variety of subgroups, including managers, employees with a manager in the treatment group, longer-tenured employees, longer-commuting employees, women, employees with children, computer engineers and those living further away, as well as looking at whether internet speed had any effect. We found no evidence of a difference in response to treatment across these groups (Extended Data Table 5).

The experiment also analysed two other measures of employee performance. First, the performance reviews at Trip.com have sub-components for individual activities such as 'innovation', 'leadership', 'development' and 'execution' (nine categories in all) when these are important for an individual employee's role. We collected these data and analysed these scores for the four six-month performance review periods. We found no evidence of a difference across these nine major categories over the four performance review periods

(Extended Data Table 6). This indicates that for categories that involve softer skills or more team-focused activities—such as development and innovation—there is no evidence for a material effect of being randomized into the hybrid WFH treatment. Second, for the 653 computer engineers, we obtained data on the lines of code uploaded by each engineer each day. For this 'lines of code submitted' measure, we found no difference between employees in the control and treatment groups (Extended Data Fig. 1 and 'Null results' section of the Methods).

## Self-assessed productivity

All experiment participants were polled before the experiment in a baseline survey on 29 and 30 July 2021, which included a two-part question on their beliefs about the effects of hybrid WFH on productivity. Employees were asked 'What is your expectation for the impact of hybrid WFH on your productivity?', with three options of 'positive', 'about the same' or 'negative'. Individuals who chose the answer 'positive' were then offered a set of options asking how positive they felt, ranging from [5% to 15%] up to [35% or more], and similarly so for negative choices. For aggregate impacts we took the mid-points of each bin, and 42.5% for >35% and −42.5% for <−35%. Employees were

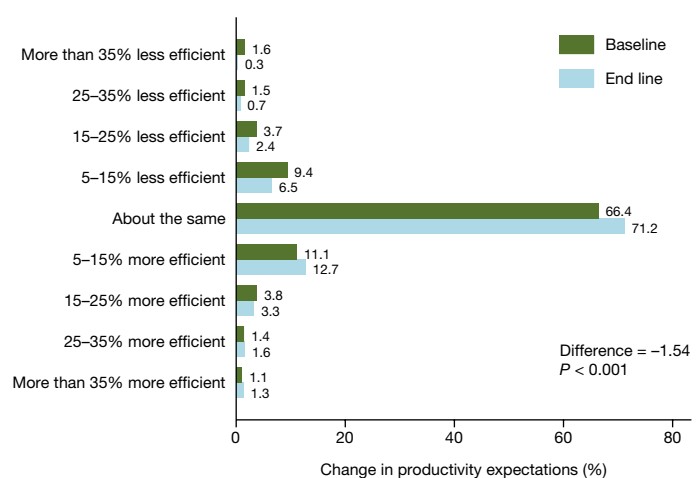

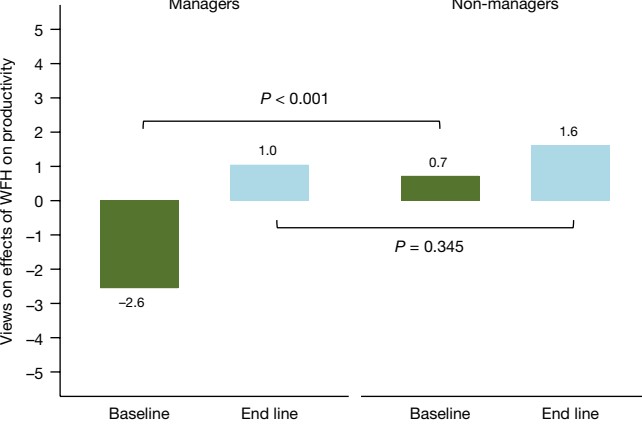

**Fig. 5 | Views on the effect of WFH on productivity improved after the experiment, particularly for managers.** Sample from 1,315 employees (314 managers, 1,001 non-managers) at the baseline and 1,345 employees (324 managers, 1,021 non-managers) at the end line. Two-tailed *t*-tests for the difference in productivity expectations between baseline and end line, after assigning a numeric value corresponding to the midpoint of the bucket,

are (baseline mean = −0.058, end-line mean = 1.48, difference = −1.54, s.e. = 0.40, CI = [−2.33, −0.753], *P* < 0.001). Two-tailed *t*-tests for the baseline difference between the productivity expectations of managers and non-managers are (difference = −3.28, s.e. = 0.72, CI = [−4.69, −1.86], *P* < 0.001), and the *t*-tests for the end-line difference are (difference = −0.571, s.e. = 0.604, CI = [−1.76, 0.615], *P* = 0.345).

resurveyed with the same question after the end of the experiment on 21 January 2022.

The left panel of Fig. 5 shows that employees' pre-experimental beliefs about WFH and productivity were extremely varied. The baseline mean was −0.1%, but with widespread variation (standard deviation of 11%). This spread should be unsurprising to anyone who has been following the active debate about the effects of remote work on productivity. At the end-line survey conducted on 21 January 2022, the mean of these beliefs had significantly increased to 1.5%, revealing that the experience of hybrid working led to a small improvement in average employee beliefs about the productivity impact of hybrid working (mean$_{baseline}$ = −0.06%, mean$_{endline}$ = 1.48%, $t(2658) = -3.84$, $P < 0.001$). This could be because hybrid WFH saves employees commuting time and is less physically tiring, and, with intermittent breaks between group time and quiet individual time, can improve performance[19–22].

The right panel of Fig. 5 shows that in the baseline survey, managers were negative about the perceived effect of hybrid work on their productivity, with a mean effect of −2.6%. Non-managers, by contrast, were significantly more positive, at +0.7% in the baseline survey (mean$_{non-manager}$ = 0.7%, mean$_{manager}$ = −2.6%, $t(1313) = -4.56$, $P < 0.001$). At the end of the experiment, the views of managers improved to 1.0%, with no evidence of a difference from the non-managers' mean value of 1.6% (mean$_{non-manager}$ = 1.62%, mean$_{manager}$ = 1.05%, $t(1343) = -0.95$, $P = 0.345$). Hence, the experiment led managers to positively update their views about how hybrid WFH affects productivity, and to more closely align with non-managers.

Of note, we saw that employees in the treatment and control groups had similar increases in self-assessed productivity (difference 0.58%, s.d. = 0.59%). Employees from four other divisions in Trip.com were also polled about the productivity impact of hybrid WFH after the end of the experiment in March 2022, with a mean estimate of +2.8% on a sample of 3,461 responses—similar to the 1.5% end line for the experimental sample. This suggests that even close exposure to hybrid WFH is sufficient for employees to change their views, consistent with previous evidence of a positive society-wide shift in perceptions about WFH productivity after the 2020 pandemic[8].

## Discussion

Once the experiment ended, the Trip.com executive committee examined the data and voted to extend the hybrid WFH policy to all employees in all divisions of the company with immediate effect. Their logic was that each quit cost the company approximately US$20,000 in recruitment and training, so a one-third reduction in attrition for the firm would generate millions of dollars in savings. This was publicly announced on 14 February 2022, with wide coverage in the Chinese media. Since then, other Chinese tech firms have adopted similar hybrid policies[23].

This highlights how, contrary to the previous causal research focused on fully remote work, which found mostly negative effects on productivity[5–7], hybrid remote work can leave performance unchanged. This suggests that hybrid working can be profitably adopted by organizations, given its effect on reducing attrition, which is estimated to cost about 50% of an individual's annual salary for graduate employees[24]. Hybrid working also offers large gains for society by providing a valuable amenity (perk) to employees, reducing commuting and easing child-care[6,25,26].

The experiment was conducted in a Chinese technology firm based in Shanghai. Although it might not be possible to replicate these results perfectly in other situations, Trip.com is a large multinational firm with global suppliers, customers and investors. Its offices are modern buildings that look similar to those in many American, Asian and European cities. Trip employees worked 8.6 h per day on average, close to the 8 h per day that is usual for US graduate employees[27]. The business had

a large drop in revenue in 2020 (see Extended Data Fig. 4), followed by roughly flat revenues through the 2021 experiment period into 2022, so this was not a period of exceptionally fast or slow growth. As such, we believe that these results— that is, the finding that allowing employees to WFH two days per week reduces quit rates and has a limited effect on performance—would probably extend to other organizations. Also, this experiment analysed the effects of working three days per week in the office and two days per week from home. So, our findings might not replicate to all other hybrid work arrangements, but we believe that they could extend to other hybrid settings with a similar number of days in the office, such as two or four days a week. We are not sure whether the results would extend to more remote settings such as one day a week (or less) in the office, owing to potential challenges around training, innovating and culture in fully remote settings.

Finally, we should point out two implications of the experimental design. First, full enrolment into hybrid schemes is important because of concerns that volunteering might be seen as a negative signal about career ambitions. The low volunteer rate among female employees, despite their high implied value (from the large reductions in quit rates observed), is particularly notable in this regard. Second, there is value in experimentation. Before the experiment, managers were net-negative in their views on the productivity impact of hybrid working, but after the experiment, their views became net-positive. This highlights the benefits of experimentation for firms to evaluate new working practices and technologies.

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

# Article

## Methods

### Location and set-up

Our experiment took place at Trip.com in Shanghai, China. In July 2021, Trip.com decided to evaluate hybrid WFH after seeing its popularity amongst US tech firms. The first step took place on 27 July 2021, when the firm surveyed 1,612 eligible engineers, marketing and finance employees in the Airfare and IT divisions about the option of hybrid WFH. They excluded interns and rookies who were in probation periods because on-site learning and mentoring are particularly important for those individuals. Trip.com chose these two divisions as representative of the firm, with a mix of employee types to assess any potentially heterogeneous impacts. About half of the employees in these divisions are technical employees, writing software code for the website, and front-end or back-end operating systems. The remainder work in business development, with tasks such as talking to airlines, travel agents or vendors to develop new services and products; in market planning and executing advertising and marketing campaigns; and in business services, dealing with a range of financial, regulatory and strategy issues. Across these groups, 395 individuals were managers and 1,217 non-managers, providing a large enough sample of both groups to evaluate their response to hybrid WFH.

### Randomization

The employees were sent an email outlining how the six-month experiment offered them the option (but not the obligation) to WFH on Wednesday and Friday. After the initial email and two follow-up reminders, a group of 518 employees volunteered. The firm randomized employees with odd birthdays—those born on the first, third, fifth and so on of the month—into eligibility for the hybrid WFH scheme starting on the week of 9 August. Those with even birthdays—born on the second, fourth, sixth and so on of the month—were not eligible, so formed the control group.

The top management at the firm was surprised at the low volunteer rate for the optional hybrid WFH scheme. They suspected that many employees were hesitating because of concerns that volunteering would be seen as a negative signal of ambition and productivity. This is not unreasonable. For example, a previous study[28] found in the US firm they evaluated that WFH employees were negatively selected on productivity. So, on 6 September, all of the remaining 1,094 non-volunteer employees were told that they were also included in the program. The odd-birthday employees were again randomized into the hybrid WFH treatment and began the experiment on the week of 13 September. In this paper we analyse the two groups together, but examining the volunteer and non-volunteer groups individually yields similar findings of reduced quit rates and no impact on performance.

### Employee characteristics and balancing tests

Figure 1 shows some pictures of employees working in the office (left side). Employees all worked in modern open-plan offices in desk groupings of four or six colleagues from the same team. By contrast, when WFH, they usually worked alone in their apartments, typically in the living room or kitchen (see Extended Data Fig. 2).

The individuals in the experimental sample are typically in their mid-30s. About two-thirds are male, all of them have a university undergraduate degree and almost one-third have a graduate degree (typically a master's degree). In addition, nearly half of the employees have children (details in Extended Data Table 1).

In Extended Data Table 7 we confirm that this sample is also balanced across the treatment and control groups, by conducting a two-sample $t$-test. The exceptions are from random variation given that the sampling was by even or odd day-of-month birthday—the control sample is 0.5 years older ($P = 0.06$), and this is presumably linked to why those in this group have 0.06% more children ($P = 0.02$) and 0.4 years more tenure ($P = 0.09$).

In Extended Data Table 3, we examine the decision to volunteer for the WFH experiment. We see that volunteers were significantly less likely to be managers (mean$_{non-volunteer}$ = 0.28, mean$_{volunteer}$ = 0.17, $t(1610) = -4.85$, $P < 0.001$) and had longer commute times (hours) (mean$_{non-volunteer}$ = 0.80, mean$_{volunteer}$ = 0.89, $t(1257) = 3.68$, $P < 0.001$). Notably, we don't find evidence of a relationship between volunteering and previous performance scores (mean$_{non-volunteer}$ = 3.81, mean$_{volunteer}$ = 3.81, $t(1580) = -0.02$, $P = 0.985$), highlighting, at least in this case, the lack of evidence for any negative (or positive) selection effects around WFH.

Extended Data Fig. 3 plots the take-up rates of WFH on Wednesday and Friday by volunteer and non-volunteer groups. We see a few notable facts. First, take-up overall was about 55% for volunteers and 40% for non-volunteers, indicating that both groups tended to WFH only one day, typically Friday, each week. At Trip.com, large meetings and product launches often happen mid-week, so Fridays are seen as a better day to WFH. Second, the take-up rate even for non-volunteers was 40%, indicating that Trip.com's suspicion that many employees did not volunteer out of fear of negative signalling was well-founded, and highlighting that amenities like WFH, holiday, maternity or paternity leave might need to be mandatory to ensure reasonable take-up rates. Third, take-up surged on Fridays before major holidays. Many employees returned to their home towns, using their WFH day to travel home on the quieter Thursday evening or Friday morning. Finally, take-up rates jumped for both treatment-group and control-group employees in late January 2022 after a case of COVID in the Shanghai headquarters. Trip.com allowed all employees at that point to WFH, so the experiment effectively ended early on Friday 21 January. The measure of an employee's daily WFH take-up excludes leave, sick leave or occasions when they cannot come to the office owing to extreme bad weather (typhoon) or to the COVID outbreak in the company.

### Null results

To interpret the main null results, we conduct null equivalence tests using the two one-sided tests (TOST) procedure in R (refs. 29,30). This test required us to specify the smallest effect size of interest (SESOI). For the results pertaining to performance review measures, we use 0.5 as the SESOI. This corresponds to half of a consecutive letter grade increase or decrease, because we had assigned numeric values to performance letter grades in increments of 1, with the lowest letter grade D being 1, and the highest letter grade A being 5. We performed equivalence tests for a two-sample Welch's $t$-test using equivalence bounds of ±0.5. The TOST procedure yielded significant results using the default alpha of 0.05 for the tests against both the upper and the lower equivalence bounds for the performance measures for July–December 2021 ($t(1504) = -10.20$, $P < 0.001$), January–June 2022 ($t(1353) = -10.57$, $P < 0.001$), July–December 2022 ($t(1299) = 10.34$, $P < 0.001$) and January–June 2023 ($t(1248) = -8.80$, $P < 0.001$). The equivalence test is therefore significant, which means we can reject the hypothesis that the true effect of the treatment on performance is larger than 0.5 or smaller than −0.5. So, we interpret the performance effects of the treatment to be actually null on the basis of the SESOI we used, as opposed to no evidence of a difference in performance.

We conducted null equivalence results for the effect of the treatment on promotions using 2 as the SESOI, corresponding to ±2 percentage points (pp) difference in promotion rates. Although we can reject the null hypothesis that the true effect of treatment on promotion is larger than 2 pp or smaller than −2 pp in January–June 2022 ($t(1376) = -2.22$, $P = 0.013$) and July–December 2022 ($t(1306) = 1.33$, $P = 0.092$), we fail to reject the null equivalence hypothesis in July–December 2021 ($t(1513) = 0.83$, $P = 0.203$) and January–June 2023 ($t(1250) = 0.98$, $P = 0.163$). Thus, we interpret the results on promotion as no evidence of a difference between promotion rates across treatment and control employees.

We also conducted the equivalence test for lines of code using 29 lines of code per day as the SESOI, which corresponds to 10% of the mean number of lines of code for the control group. We arrive at this SESOI on the basis of rounding down the productivity effects of previous findings[8,10]. We can reject the equivalence null hypothesis for lines of code ($t(92362) = -2.74$, $P = 0.003$)) so we interpret the effect of the treatment as a null effect.

### Volunteer versus non-volunteer groups

In the main paper we pool the volunteer and non-volunteer groups. In Extended Data Table 5 we examine the impacts on performance and promotions and we see no evidence of a difference in performance and promotion treatment effects for volunteer versus non-volunteer groups (column 9).

### Performance subcategories

The company has a rigorous performance-reviewing process every six months that determines employees' pay and promotion, so is carefully conducted. The review process for each employee is built on formal reviews provided by their managers, project leaders and sometimes co-workers (peer review). Managers are more like an employee's direct managers for organizational purposes, but for a particular project, the project leader could be another higher-level employee. In such a case, the manager of the employee would ask that project leader for an opinion on the employee's contribution to the project. An individual's overall score is a weighted sum of scores from various subcategories that managers have broad flexibility over defining, because tasks differ across employees, and managers would give a score for each task. For example, an employee running a team themselves will have subcategories around developing their direct reports (leadership and communication), whereas an employee running a server network will have subcategories around efficiency and execution. The performance subcategory data come from the text of the performance review. We first used the most popular Chinese word segmentation package in Python, named Jieba, to identify the most frequent Chinese words from task titles across four performance reviews. We also removed meaningless words and incorporated common expressions such as key performance indicators ('KPI'), objectives and key results ('OKR'), 'rate' and '%'. This process resulted in a total of 236 unique words and expressions. We then manually categorized those most frequent keywords into nine major subcategories (see below) by meanings and relevance. Finally, on the basis of the presence of keywords in the task title, tasks were grouped into the following subcategories:

- Communication tasks are those that involve communication, collaboration, cooperation, coordination, participation, suggestion, assistance, organization, sharing and relationships.
- Development tasks are those that involve coding or codes, data or datasets, systems, techniques and skills.
- Efficiency tasks are those that involve cost reduction, ratios, return on investment (ROI), rate, %, improvement, growth, lifting, adding, optimizing, profit, receiving, gross merchandise value (GMV), OKR, KPI, work and goal.
- Execution tasks are those that involve execution, conducting, maintenance, delivery, output, quality, contribution and workload.
- Innovation tasks are those that involve development, R&D and innovation.
- Leadership tasks are those that involve leadership, managing or management, approval, internal, strategy, coordination and planning.
- Learning tasks are those that involve learning, growing, maturing, talent, ability, value competitiveness and personal improvement.
- Project tasks are those that involve project, supply, product, business line, cooperation and clients.
- Risk tasks are those that involve risk, compliance, supervision, recording and monitoring, safety, rules and privacy.

### Data sources

Data were provided by a combination of Trip.com sources, including human resources records, performance reviews and two surveys. All data were anonymized and coded using a scrambled individual ID code, so no personally identifiable information was shared with the Stanford team. The data were drawn directly from the Trip.com administrative data systems on a monthly basis. Gender is collected by Trip.com from employees when they join the company.

### Subsamples

The full sample has 1,612 experiment participants, but we have 1,507, 1,355, 1,301 and 1,254 employees, respectively, in the subsamples for the four performance reviews from July–December 2021, January–June 2022, July–December 2022 and January–June 2023. These smaller samples are due to attrition. In addition, for the first performance review in July–December 2021, 105 employees did not have sufficient pre-experiment tenure to support a performance review (they had joined the firm less than three months before the experimental draw). The review text data covers 1,507, 1,339, 1,290 and 1,246 people, as some employees do have an overall score and review text but do not have additional and task-specific scores. The reason is that these employees do not have the full range of all tasks, so their managers did not write the full review script. For the two surveys, Trip.com used Starbucks vouchers to incentivize response and collected responses from 1,315 employees (314 managers, 1,001 non-managers) at the baseline on the left, and that of 1,345 employees (324 managers, 1,021 non-managers) at the end line.

### Testings

All tests used two-sided Student $t$-tests unless otherwise stated. Analysis was run on Stata v17 and v18, R version 4.2.2. Unless stated otherwise, no additional covariates are included in the tests. The null hypothesis for all of the tests excluding null equivalence tests is a coefficient of zero (for example, zero difference between treatment and control).

### Inclusion and ethics statement

The design and execution of the experiment was run by Trip.com. No participants were forced to WFH owing to the experiment (the entire firm was, however, forced to WFH during the pandemic lockdown). The treatment sample had the option but not the obligation to WFH on Wednesday or Friday. The experiment was designed, initiated and run by Trip.com. N.B. and R.H. were invited to analyse the data from the experiment, with consent for data collection coming from Trip.com internally. The experiment was exempt under institutional review board (IRB) approval guidelines because it was designed and initiated by Trip.com, before N.B. and R.H. were invited to analyse the data. Only anonymous data were shared with the Stanford team. Trip.com based the experimental design and execution on their previous experience with WFH randomized control trials[17].

### Reporting summary

Further information on research design is available in the Nature Portfolio Reporting Summary linked to this article.

### Data availability

The data necessary to reproduce the primary results of this study can be found at https://doi.org/10.7910/DVN/6X4ZZL. These data have been anonymized and split into individual files to ensure that no individual is identifiable. All figures and tables can be replicated using this data.

### Code availability

The code necessary to reproduce the primary results of this study can be found at https://doi.org/10.7910/DVN/6X4ZZL.

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

**Acknowledgements** We thank the Smith Richardson Foundation for funding; J. Cao, T. Zhang, S. Ye, F. Chen, X. Zhang, Y. He, J. Li, B. Ye and M. Akan for data, advice and logistical support; D. Yilin for research assistance; S. Ayan, S. Buckman, S. Gurung, M. Jackson and P. Lambert for draft feedback; and J. Sun for project leadership.

**Author contributions** N.B. oversaw the analysis, presented the results and wrote the main drafts of the paper. He was the principal investigator on the research grant supporting the research. R.H. supervised data collection and analysed the data, presented the results and helped to draft the paper. J.L. initiated and designed the study, discussed the results and analysis and facilitated the Trip.com engagement. N.B. and R.H. are co-first authors.

**Competing interests** No funding was received from Trip.com. J.L. is the co-founder, former CEO and current chairman of Trip.com, with equity holdings in Trip.com. No other co-author has any financial relationship with Trip.com. Neither the results nor the paper was pre-screened by anyone. The experiment was registered with the American Economic Association on 16 August 2021 after the experiment had begun but before N.B. and R.H. had received any data. Only anonymous data were shared with the Stanford team.

**Additional information**
**Correspondence and requests for materials** should be addressed to Nicholas Bloom, Ruobing Han or James Liang.

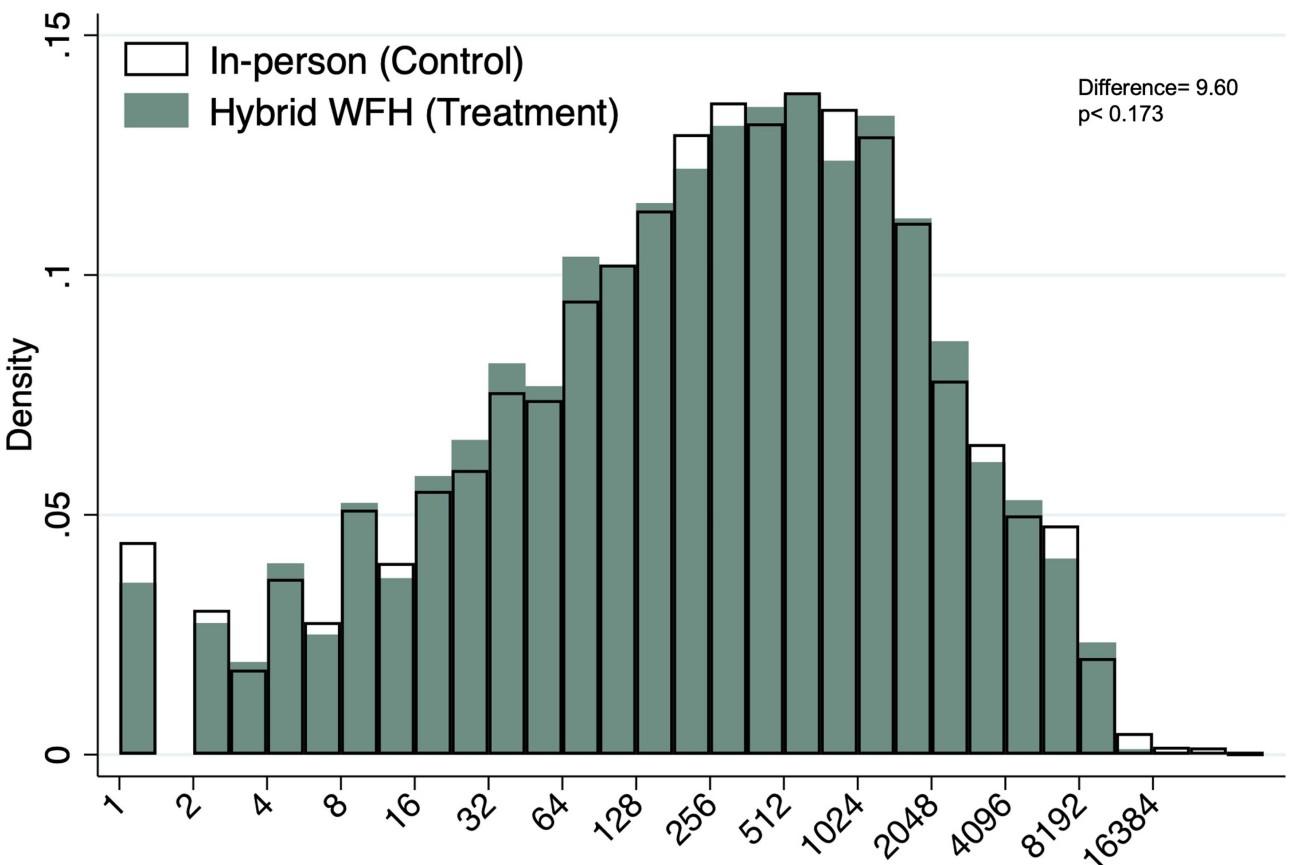

**Extended Data Fig. 1 | WFH had no effect on lines of code written.** The data coves the experimental period starting on 9 August 2021 for the first wave and 13 September for the second wave, running to 23 January 2022, for both waves. Lines of code submitted per day is available for 653 employees whose primary role was writing code, spanning a total of 95,494 days. Lines are those uploaded to trip.com on a daily basis. Data plotted on a log-2 scale for readability. Reported $P$ value is calculated using a two-sided $t$-test on the number of code lines and the difference is for control minus treatment. When using $\log_2$(code lines) the difference has a $P$ value of 0.750 (noting the sample is 27,605 days because of dropping 0 values). When using $\log_2$(1 + code lines) the difference has a $P$ value of 0.0103, with treatment having the higher average values. The null equivalence tests are included in the 'Null results' section of the Methods.

# Article

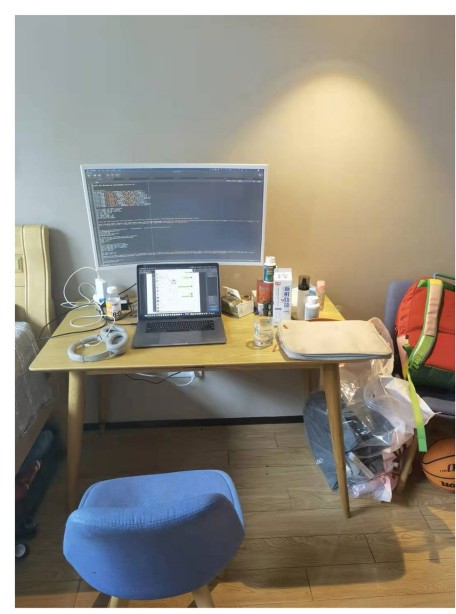
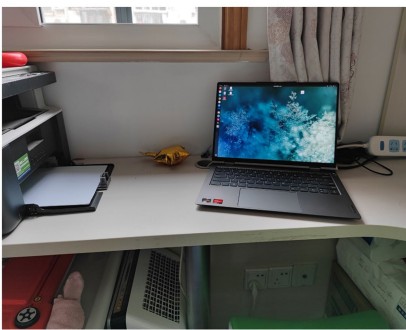
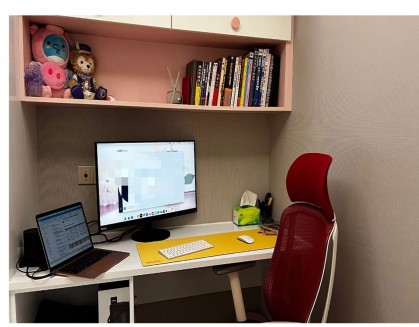
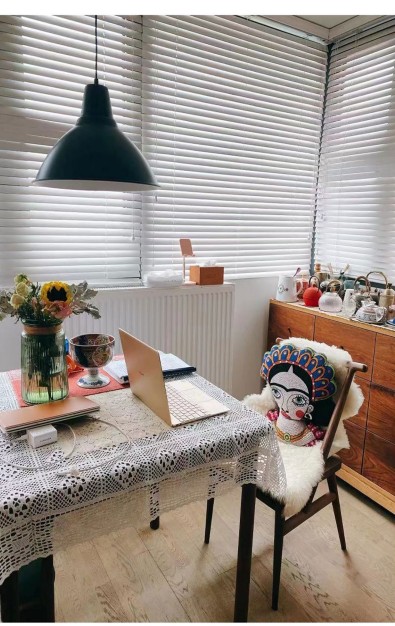

**Extended Data Fig. 2 | Home (October 2021).** Employees set up basic working environments in their living rooms, studies, or kitchens, and bring back company laptops if necessary.

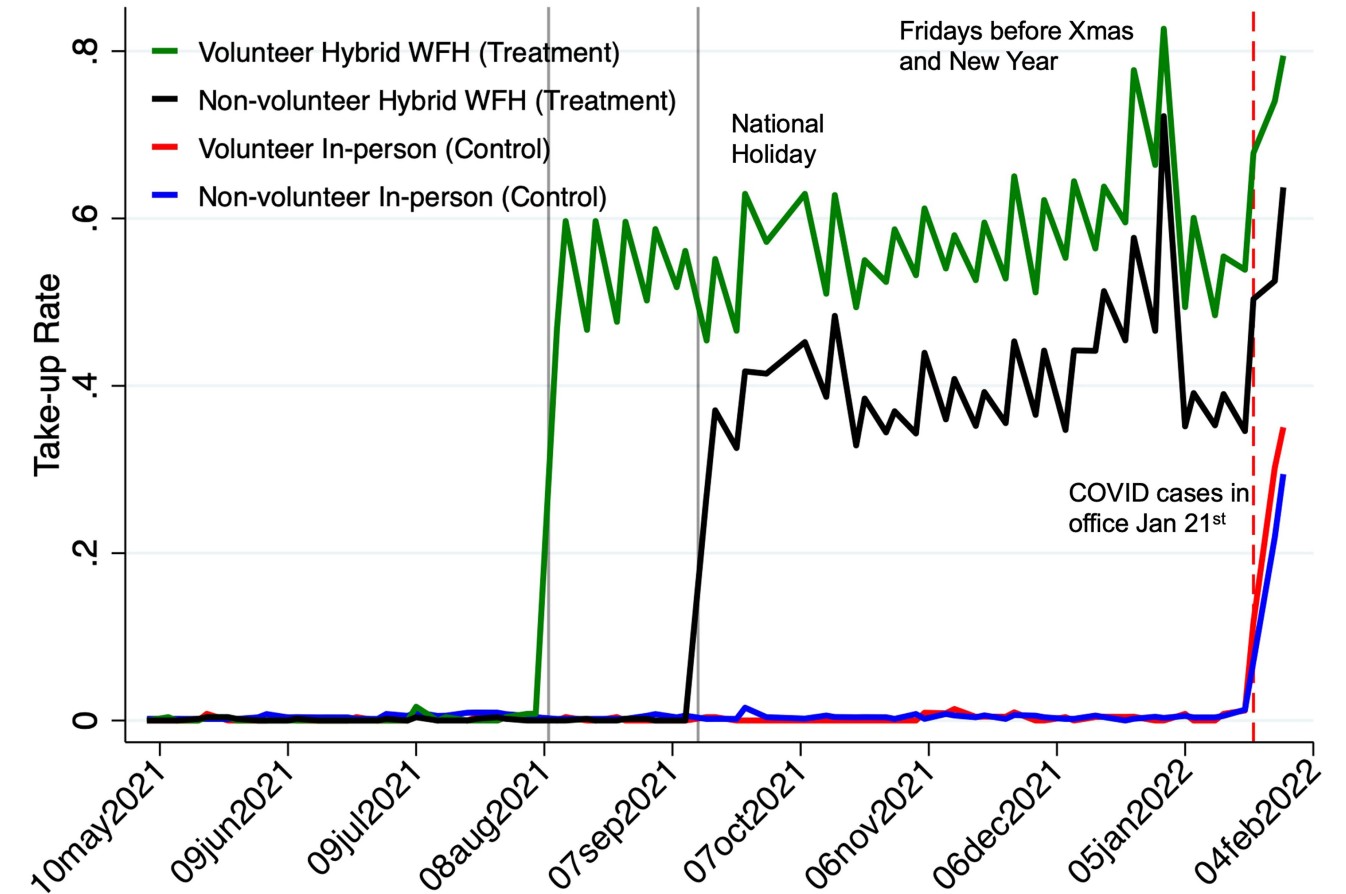

**Extended Data Fig. 3 | Take-up rate for WFH treatment and control by volunteer status.** Data for 1,612 employees from 9 August 2021 (volunteers) and 13 September (non-volunteers) to 23 January 2022. Public holidays, personal holidays and excused absence (for example, sick leave) are excluded. Take-up rate is percentage of Wednesday and Friday each week they WFH.

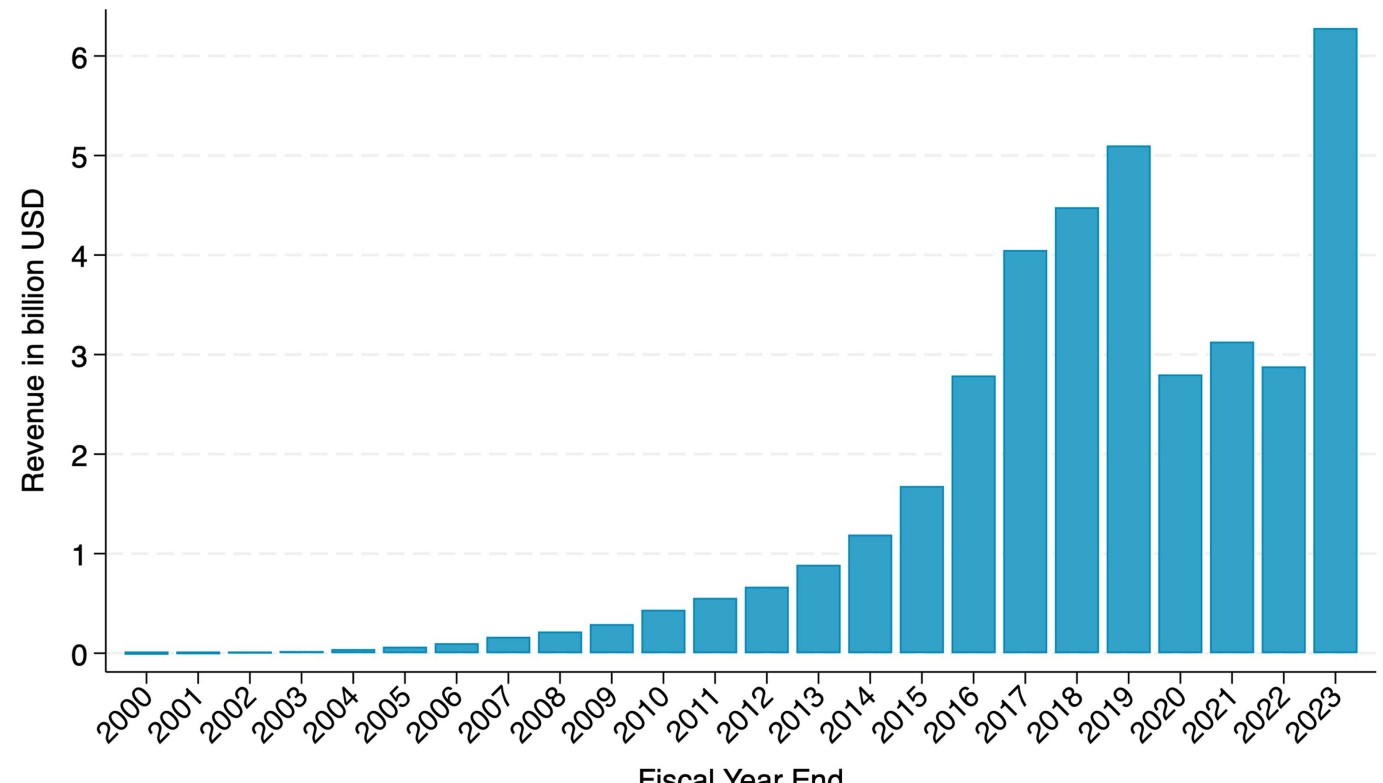

**Extended Data Fig. 4 | Trip.com revenues.** Trip.com revenues from 2000 to 2023.

**Extended Data Table 1 | Descriptive statistics**

|  | n | mean | sd | min | p10 | median | p90 | max |
|---|---|---|---|---|---|---|---|---|
| Age | 1612 | 32.61 | 5.40 | 22 | 26 | 32 | 40 | 52 |
| Children | 1612 | 0.48 | 0.50 | 0 | 0 | 0 | 1 | 1 |
| Commute (hours) | 1259 | 0.83 | 0.43 | 0.033 | 0.33 | 0.75 | 1.50 | 2 |
| Female | 1612 | 0.35 | 0.48 | 0 | 0 | 0 | 1 | 1 |
| Manager | 1612 | 0.25 | 0.43 | 0 | 0 | 0 | 1 | 1 |
| Post Graduate Degree | 1612 | 0.32 | 0.47 | 0 | 0 | 0 | 1 | 1 |
| Prior Performance | 1582 | 3.81 | 0.76 | 1 | 3 | 4 | 5 | 5 |
| Tenure (years) | 1612 | 6.35 | 4.63 | 0.33 | 1.33 | 5.75 | 13.8 | 22 |
| Hybrid WFH (Treatment) | 1612 | 0.49 | 0.50 | 0 | 0 | 0 | 1 | 1 |

Data from 1,612 experiment participants. Commute (hours) is one way. Only 1,259 employees filled out the baseline survey question on commuting length. Only 1,582 employees had a previous performance review, with the other 30 employees having joined too recently to have this.

**Extended Data Table 2 | Job-satisfaction-survey measures were higher for employees in the treatment group**

|  | In-person (Control) | Hybrid WFH (Treatment) | Diff. | p-value | n |
|---|---|---|---|---|---|
| Work-life balance | 6.946 | 7.484 | 0.538 | 0.000 | 1345 |
| Work satisfaction | 7.835 | 8.185 | 0.351 | 0.000 | 1345 |
| Life satisfaction | 7.462 | 7.813 | 0.352 | 0.000 | 1345 |
| Recommend to friends | 7.994 | 8.346 | 0.352 | 0.000 | 1345 |
| Intention to quit | -0.107 | -0.131 | -0.024 | 0.008 | 1345 |

Sample from 1345 employees (446 volunteers, 899 non-volunteers) in the end-line survey. Values for intention to quit range from −0.325 (lowest) to 0.325 (highest). Values for the other variables range from 0 (lowest) to 10 (highest). For example, 'Would recommend to friends' ranges from 'Definitely no' at 0 to 'Definitely yes' at 10. $P$ values are calculated using two-tailed $t$-tests for the difference between the control and treatment groups.

**Extended Data Table 3 | Volunteers are more likely to be non-managers, have longer commutes and less tenure**

|  | Non-volunteer Obs | Non-volunteer Mean | Volunteer Obs | Volunteer Mean | Diff. | p-value |
|---|---|---|---|---|---|---|
| Age | 1094 | 32.92 | 518 | 31.96 | -0.96 | 0.00 |
| Children | 1094 | 0.49 | 518 | 0.46 | -0.03 | 0.33 |
| Commute (hours) | 816 | 0.80 | 443 | 0.89 | 0.09 | 0.00 |
| Female | 1094 | 0.37 | 518 | 0.32 | -0.05 | 0.04 |
| Manager | 1094 | 0.28 | 518 | 0.17 | -0.11 | 0.00 |
| Post Graduate Degree | 1094 | 0.32 | 518 | 0.34 | 0.02 | 0.48 |
| Prior Performance | 1071 | 3.81 | 511 | 3.81 | 0.00 | 0.99 |
| Tenure (years) | 1094 | 6.84 | 518 | 5.31 | -1.53 | 0.00 |

Data from 1,612 experiment participants. Commute (hours) is one way. Only 1,259 employees filled out the baseline survey question on commuting length. Only 1,582 employees had a previous performance review, with the other 30 employees having joined too recently to have this. *P* values are calculated using two-tailed *t*-tests for the difference between volunteer and non-volunteer groups.

**Extended Data Table 4 | Tests of the productivity impact of hybrid WFH**

| | In-person (Control) Obs | In-person Mean | Hybrid WFH (Treatment) Obs | Hybrid WFH Mean | Diff. | SE | p-value |
|---|---|---|---|---|---|---|---|
| Performance 2021 Jul-Dec | 759 | 3.49 | 748 | 3.44 | -0.06 | 0.04 | 0.20 |
| Performance 2022 Jan-Jun | 681 | 3.55 | 674 | 3.51 | -0.03 | 0.04 | 0.44 |
| Performance 2022 Jul-Dec | 655 | 3.50 | 646 | 3.52 | 0.02 | 0.05 | 0.68 |
| Performance 2023 Jan-Jun | 633 | 3.46 | 621 | 3.41 | -0.05 | 0.05 | 0.37 |
| Promotion 2021 Jul-Dec | 766 | 6.92 | 756 | 7.80 | 0.89 | 1.34 | 0.51 |
| Promotion 2022 Jan-Jun | 693 | 2.60 | 685 | 2.48 | -0.12 | 0.85 | 0.89 |
| Promotion 2022 Jul-Dec | 661 | 4.08 | 653 | 4.59 | 0.51 | 1.12 | 0.65 |
| Promotion 2023 Jan-Jun | 647 | 2.94 | 636 | 3.93 | 0.99 | 1.02 | 0.33 |
| Lines of code | 46767 | 289.97 | 48727 | 280.37 | -9.60 | 7.05 | 0.17 |

Sample sizes for promotions, 1,522 employees in July to December 2021, 1,378 employees in January to June 2022, 1,314 employees in July to December 2022 and 1,283 employees in January to June 2023; for performance reviews, 1,507 employees in July to December 2021, 1,355 employees in January to June 2022, 1,301 employees in July to December 2022 and 1,254 employees in January to June 2023; and 653 employees for lines of code. All tests two-sided $t$-tests of a zero-null. The null equivalence tests are included in the 'Null results' section of the Methods.

**Extended Data Table 5 | No evidence of performance or promotion treatment heterogeneity**

Left table (dependent variable: performance measure):

| | (1) | (2) | (3) | (4) | (5) | (6) | (7) | (8) | (9) |
|---|---|---|---|---|---|---|---|---|---|
| Hybrid WFH | -0.056 {0.198} | 0.011 {0.860} | -0.070 {0.302} | -0.032 {0.602} | -0.075 {0.158} | -0.066 {0.177} | -0.030 {0.614} | -0.047 {0.433} | -0.064 {0.223} |
| Children | | -0.043 {0.477} | | | | | | | |
| Hybrid WFH*Children | | -0.151 {0.079} | | | | | | | |
| Long commute | | | -0.129 {0.061} | | | | | | |
| Hybrid WFH*Long commute | | | 0.029 {0.768} | | | | | | |
| Engineer | | | | -0.078 {0.200} | | | | | |
| Hybrid WFH*Engineer | | | | -0.051 {0.554} | | | | | |
| Female | | | | | -0.127 {0.048} | | | | |
| Hybrid WFH*Female | | | | | 0.064 {0.483} | | | | |
| Manager | | | | | | 0.286 {0.000} | | | |
| Hybrid WFH*Manager | | | | | | 0.058 {0.552} | | | |
| Long tenure | | | | | | | -0.133 {0.028} | | |
| Hybrid WFH*Long tenure | | | | | | | -0.065 {0.448} | | |
| Hybrid WFH manager | | | | | | | | 0.043 {0.483} | |
| Hybrid WFH*Hybrid WFH manager | | | | | | | | -0.021 {0.810} | |
| Volunteer | | | | | | | | | -0.081 {0.219} |
| Hybrid WFH*Volunteer | | | | | | | | | 0.029 {0.757} |
| N | 1507 | 1507 | 1189 | 1507 | 1507 | 1507 | 1507 | 1507 | 1507 |

Right table (dependent variable: promotion indicator):

| | (1) | (2) | (3) | (4) | (5) | (6) | (7) | (8) | (9) |
|---|---|---|---|---|---|---|---|---|---|
| Hybrid WFH | 0.009 {0.503} | -0.007 {0.723} | -0.004 {0.838} | -0.003 {0.880} | 0.020 {0.224} | -0.000 {0.992} | 0.014 {0.460} | 0.024 {0.193} | 0.011 {0.518} |
| Children | | -0.081 {0.000} | | | | | | | |
| Hybrid WFH*Children | | 0.023 {0.389} | | | | | | | |
| Long commute | | | -0.052 {0.018} | | | | | | |
| Hybrid WFH*Long commute | | | 0.028 {0.367} | | | | | | |
| Engineer | | | | 0.051 {0.008} | | | | | |
| Hybrid WFH*Engineer | | | | 0.025 {0.344} | | | | | |
| Female | | | | | 0.007 {0.711} | | | | |
| Hybrid WFH*Female | | | | | -0.032 {0.261} | | | | |
| Manager | | | | | | -0.067 {0.002} | | | |
| Hybrid WFH*Manager | | | | | | 0.034 {0.274} | | | |
| Long tenure | | | | | | | -0.058 {0.002} | | |
| Hybrid WFH*Long tenure | | | | | | | -0.015 {0.568} | | |
| Hybrid WFH manager | | | | | | | | 0.036 {0.063} | |
| Hybrid WFH*Hybrid WFH manager | | | | | | | | -0.034 {0.213} | |
| Volunteer | | | | | | | | | -0.003 {0.903} |
| Hybrid WFH*Volunteer | | | | | | | | | -0.005 {0.875} |
| N | 1507 | 1507 | 1189 | 1507 | 1507 | 1507 | 1507 | 1507 | 1507 |

Regression result tables, with each column being OLS regression. The dependent variables are the performance measure after assigning each letter grade a numeric value from 1 (D) to 5 (A) (left) and promotion indicator (right). Children is an indicator for having children and long commute is an indicator for commuters with above-median commute time, which is missing in unreported cases. All others are binary definitions. $P$ values for two-tailed $t$-tests are reported in curly brackets. Sample sizes in the row 'N' at the foot of the table may be lower than the full sample if data are only present for subsamples (for example, survey respondents). No other covariates are included.

**Extended Data Table 6 | No evidence of a difference on performance subcategories for the WFH group**

| | Communication | Development | Efficiency | Execution | Innovation | Leadership | Learning | Project | Risk |
|---|---|---|---|---|---|---|---|---|---|
| **2021 July-December** | | | | | | | | | |
| Hybrid WFH (Treatment) Mean | 3.73 | 3.73 | 3.74 | 3.72 | 3.73 | 3.74 | 3.73 | 3.73 | 3.76 |
| In-person (Control) Mean | 3.70 | 3.71 | 3.73 | 3.72 | 3.74 | 3.71 | 3.70 | 3.71 | 3.75 |
| P-value of difference | 0.44 | 0.51 | 0.54 | 0.99 | 0.70 | 0.51 | 0.40 | 0.46 | 0.92 |
| N | 1036 | 1093 | 1479 | 1408 | 853 | 608 | 1148 | 1372 | 1018 |
| **2022 January-June** | | | | | | | | | |
| Hybrid WFH (Treatment) Mean | 3.74 | 3.72 | 3.72 | 3.71 | 3.70 | 3.70 | 3.71 | 3.71 | 3.74 |
| In-person (Control) Mean | 3.75 | 3.75 | 3.76 | 3.76 | 3.74 | 3.71 | 3.74 | 3.75 | 3.75 |
| P-value of difference | 0.71 | 0.33 | 0.10 | 0.044 | 0.27 | 0.82 | 0.44 | 0.12 | 0.72 |
| N | 959 | 973 | 1307 | 1262 | 794 | 552 | 1058 | 1240 | 948 |
| **2022 July-December** | | | | | | | | | |
| Hybrid WFH (Treatment) Mean | 3.72 | 3.73 | 3.75 | 3.72 | 3.72 | 3.68 | 3.72 | 3.73 | 3.73 |
| In-person (Control) Mean | 3.73 | 3.72 | 3.74 | 3.72 | 3.74 | 3.71 | 3.74 | 3.72 | 3.73 |
| P-value of difference | 0.80 | 0.64 | 0.60 | 0.91 | 0.59 | 0.51 | 0.43 | 0.64 | 0.78 |
| N | 910 | 935 | 1257 | 1199 | 765 | 521 | 996 | 1196 | 906 |
| **2023 January-June** | | | | | | | | | |
| Hybrid WFH (Treatment) Mean | 3.78 | 3.78 | 3.77 | 3.77 | 3.79 | 3.78 | 3.74 | 3.77 | 3.78 |
| In-person (Control) Mean | 3.75 | 3.74 | 3.78 | 3.75 | 3.78 | 3.78 | 3.73 | 3.76 | 3.78 |
| P-value of difference | 0.50 | 0.17 | 0.86 | 0.61 | 0.78 | 0.91 | 0.86 | 0.66 | 0.85 |
| N | 845 | 872 | 1188 | 1144 | 711 | 481 | 900 | 1156 | 808 |

Results from 1,507 employees in July to December 2021, 1,339 employees in January to June 2022, 1,290 employees in July to December 2022 and 1,246 employees in January to June 2023. Probability values for difference in treatment and control distributions calculated using two-sided $t$-tests. Results reported for subcategories in which a score existed—so, for example, in July to December 2021, 1,093 employees were assessed for 'Development'.

**Extended Data Table 7 | Balance table**

|  | In-person (Control) | | | Hybrid WFH (Treatment) | | | | |
|---|---|---|---|---|---|---|---|---|
|  | n | mean | sd | n | mean | sd | Diff. | p-value |
| Age | 820 | 32.86 | 5.47 | 792 | 32.35 | 5.31 | -0.515 | 0.06 |
| Children | 820 | 0.51 | 0.50 | 792 | 0.45 | 0.50 | -0.059 | 0.02 |
| Commute (hours) | 636 | 0.82 | 0.43 | 623 | 0.84 | 0.43 | 0.024 | 0.33 |
| Female | 820 | 0.35 | 0.48 | 792 | 0.36 | 0.48 | 0.017 | 0.47 |
| Manager | 820 | 0.25 | 0.43 | 792 | 0.24 | 0.43 | -0.005 | 0.81 |
| Post Graduate Degree | 820 | 0.31 | 0.46 | 792 | 0.34 | 0.47 | 0.034 | 0.15 |
| Prior Performance | 808 | 3.79 | 0.78 | 774 | 3.83 | 0.74 | 0.035 | 0.36 |
| Tenure (years) | 820 | 6.54 | 4.66 | 792 | 6.15 | 4.58 | -0.396 | 0.09 |

Data from 1,612 experiment participants. Commute (hours) is one way. Only 1,259 employees filled out the baseline survey question on commuting length. Only 1,582 employees had a previous performance review, with the other 30 employees having joined too recently to have this. Control and treatment randomization was by odd–even birthday within the month. P values are calculated using two-tailed t-tests for the difference between the control and treatment groups.

# Reporting Summary

## Statistics

For all statistical analyses, confirm that the following items are present in the figure legend, table legend, main text, or Methods section.

| n/a | Confirmed | |
|---|---|---|
| ☐ | ☒ | The exact sample size (*n*) for each experimental group/condition, given as a discrete number and unit of measurement |
| ☒ | ☐ | A statement on whether measurements were taken from distinct samples or whether the same sample was measured repeatedly |
| ☐ | ☒ | The statistical test(s) used AND whether they are one- or two-sided<br>*Only common tests should be described solely by name; describe more complex techniques in the Methods section.* |
| ☐ | ☒ | A description of all covariates tested |
| ☐ | ☒ | A description of any assumptions or corrections, such as tests of normality and adjustment for multiple comparisons |
| ☐ | ☒ | A full description of the statistical parameters including central tendency (e.g. means) or other basic estimates (e.g. regression coefficient) AND variation (e.g. standard deviation) or associated estimates of uncertainty (e.g. confidence intervals) |
| ☐ | ☒ | For null hypothesis testing, the test statistic (e.g. *F*, *t*, *r*) with confidence intervals, effect sizes, degrees of freedom and *P* value noted<br>*Give P values as exact values whenever suitable.* |
| ☒ | ☐ | For Bayesian analysis, information on the choice of priors and Markov chain Monte Carlo settings |
| ☒ | ☐ | For hierarchical and complex designs, identification of the appropriate level for tests and full reporting of outcomes |
| ☐ | ☒ | Estimates of effect sizes (e.g. Cohen's *d*, Pearson's *r*), indicating how they were calculated |

*Our web collection on statistics for biologists contains articles on many of the points above.*

## Software and code

Policy information about availability of computer code

| Data collection | Data collected by Trip.com internally. |
|---|---|
| Data analysis | Stata 17 and 18, R version 4.2.2 |

For manuscripts utilizing custom algorithms or software that are central to the research but not yet described in published literature, software must be made available to editors and reviewers. We strongly encourage code deposition in a community repository (e.g. GitHub). See the Nature Portfolio guidelines for submitting code & software for further information.

## Data

Policy information about availability of data

All manuscripts must include a data availability statement. This statement should provide the following information, where applicable:
- Accession codes, unique identifiers, or web links for publicly available datasets
- A description of any restrictions on data availability
- For clinical datasets or third party data, please ensure that the statement adheres to our policy

Data necessary to reproduce the primary results of this study are included in www.bit.ly/3qS477O This data has been anonymized and split into individual files to ensure no individual is identifiable. All figures and tables can be replicated using this data.

# Research involving human participants, their data, or biological material

Policy information about studies with human participants or human data. See also policy information about sex, gender (identity/presentation), and sexual orientation and race, ethnicity and racism.

| | |
|---|---|
| Reporting on sex and gender | Gender is collected by Trip.com from employees when they join the company. |
| Reporting on race, ethnicity, or other socially relevant groupings | N/A |
| Population characteristics | N/A |
| Recruitment | N/A |
| Ethics oversight | N/A |

Note that full information on the approval of the study protocol must also be provided in the manuscript.

# Field-specific reporting

Please select the one below that is the best fit for your research. If you are not sure, read the appropriate sections before making your selection.

☐ Life sciences    ☒ Behavioural & social sciences    ☐ Ecological, evolutionary & environmental sciences

For a reference copy of the document with all sections, see nature.com/documents/nr-reporting-summary-flat.pdf

# Behavioural & social sciences study design

All studies must disclose on these points even when the disclosure is negative.

| | |
|---|---|
| Study description | 6-month randomized-control trial of hybrid working from home. |
| Research sample | 1612 employees in a Chinese technology company in 2021-2022. |
| Sampling strategy | All engineering, marketing and finance employees in the Airfare and IT divisions excluding interns and new hires. |
| Data collection | Data collected by the company internally. |
| Timing | Experiment started ran from July 2021 to January 2022. |
| Data exclusions | No data were excluded. |
| Non-participation | All employees (excluding interns and new hires) in the Airfare and IT divisions participated in the experiment. |
| Randomization | The firm randomized with a coin flip that odd employees, those born on the 1st, 3rd, 5th, 7th, etc., would become eligible for the hybrid-WFH scheme starting on the week of the 9th of August.  Those with even birthday, so born on the 2nd, 4th, 6th etc. of the month would not be eligible so form the control group. |

# Reporting for specific materials, systems and methods

We require information from authors about some types of materials, experimental systems and methods used in many studies. Here, indicate whether each material, system or method listed is relevant to your study. If you are not sure if a list item applies to your research, read the appropriate section before selecting a response.

## Materials & experimental systems

| n/a | Involved in the study |
|-----|----------------------|
| ☒ ☐ | Antibodies |
| ☒ ☐ | Eukaryotic cell lines |
| ☒ ☐ | Palaeontology and archaeology |
| ☒ ☐ | Animals and other organisms |
| ☒ ☐ | Clinical data |
| ☒ ☐ | Dual use research of concern |
| ☒ ☐ | Plants |

## Methods

| n/a | Involved in the study |
|-----|----------------------|
| ☒ ☐ | ChIP-seq |
| ☒ ☐ | Flow cytometry |
| ☒ ☐ | MRI-based neuroimaging |

## Plants

Seed stocks — *Report on the source of all seed stocks or other plant material used. If applicable, state the seed stock centre and catalogue number. If plant specimens were collected from the field, describe the collection location, date and sampling procedures.*

Novel plant genotypes — *Describe the methods by which all novel plant genotypes were produced. This includes those generated by transgenic approaches, gene editing, chemical/radiation-based mutagenesis and hybridization. For transgenic lines, describe the transformation method, the number of independent lines analyzed and the generation upon which experiments were performed. For gene-edited lines, describe the editor used, the endogenous sequence targeted for editing, the targeting guide RNA sequence (if applicable) and how the editor was applied.*

Authentication — *Describe any authentication procedures for each seed stock used or novel genotype generated. Describe any experiments used to assess the effect of a mutation and, where applicable, how potential secondary effects (e.g. second site T-DNA insertions, mosiacism, off-target gene editing) were examined.*

