## [Peer Review File · Nature]

Manuscript Title: Hybrid Working from Home Improves Retention Without Damaging Performance

Editorial Notes:

Reviewer Comments & Author Rebuttals

Reviewer Reports on the Initial Version:

Referees' comments:

Referee #1 (Remarks to the Author):

1. The authors are working on a timely and important topic. Their results have immediate policy relevance for firms around the world.
2. In the abstract and introduction, the authors should highlight that the effects are only statistically significant for females, non-managers and workers with commute > 90mins (highlighting the null effects reported in Figure 2)
3. Given that 48% employees have children, is their variation in effects for parents and non-parents?
4. Can the authors provide a monetary value for the 33% lower attrition (i.e., savings in costs of hiring and training?)
5. Clarify if the work arrangements went back to fully in-person or continued hybrid after 6 months and how this affects the interpretation of performance results?
6. Extended Figure 5 could show take-up rate by gender given the third policy implication mentioned in page 9?
7. The final sentence in the abstract ("Three days per week in the office appears to be provide sufficient in-person interaction") could be edited given that the authors do not study the effect of two days in office or one day in office?

Referee #2 (Remarks to the Author):

I appreciate the opportunity to read the manuscript entitled "Hybrid Working from Home Improves Employee Retention Without Damaging Performance" submitted to Nature. The current study conducted a field experiment to test the effects of hybrid work policy on attrition rate, satisfaction, and work performance. The study features several strengths, such as a field study with randomization, clear data analysis process, and logical arguments for the focal links.

As with any manuscript, I identified a couple of issues that require your attention before moving on with this line of research. Below, I outline several critical major aspects that may or may not be fixed straightforwardly. I hope my comments will be helpful to you.

1. Contribution

First of all, I would like to express my concern regarding whether and how this research really moves the needle in terms of advancing our society and organizational science (e.g., management and

organizational psychology) in this area. Although I found the ideas presented to be better argued in general, I still did not find the case for the significance of the study (over and above prior work) to be sufficient. I generally agreed because the current study's findings either mirror the existing literature or could easily be intuited including their previous paper in 2015. For example, several recent papers from the management and organizational behavioral science found the benefits of hybrid work and WFH on employees' satisfaction as well as performance (e.g., Gajendran & Harrison, 2007; Gibson, Gilson, Griffith, O'Neill, 2023; Teng-Calleja, Mactal, Caringal-Go, 2023; Wood et al., 2022). Overall, I am concerned about how the current theory paper advances the literature and theory above and beyond prior research on eating and exercise.

- Gajendran, R. S., & Harrison, D. A. (2007). The good, the bad, and the unknown about telecommuting: meta-analysis of psychological mediators and individual consequences. *Journal of applied psychology*, 92(6), 1524-1541.
- Gibson, C. B., Gilson, L. L., Griffith, T. L., & O'Neill, T. A. (2023). Should employees be required to return to the office?. *Organizational Dynamics*, 52(2), 100981
- Teng-Calleja, M., Mactal, M. T. D. G., & Caringal-Go, J. F. (2023). Examining employee experiences of hybrid work: an ecological approach. *Personnel Review*.
- Wood, S., Michaelides, G., Inceoglu, I., Niven, K., Kelleher, A., Hurren, E., & Daniels, K. (2022). Satisfaction with one's job and working at home in the COVID-19 pandemic: A two-wave study. *Applied Psychology*.

2. Study Context.

The authors explained the study period from the second half of 2021 to the beginning of 2022 in Shanghai. As we know, Chinese policy on COVID-19 was rigorous and controlled many social functions (E.g., quarantine, lockdown, and full inspection of COVID-19). I was curious about the representativeness of the context to the normal period or endemic society. In other words, the practice of WFH and its impacts might be influenced by other contextual characteristics. Different from their previous study (Bloom, Liang, Roberts, & Ying, 2015), it is difficult to argue that the study period represents the normality. Additionally, the perception and motivation for hybrid work and WFH have changed since the COVID-19 outbreak. To make a significant scientific implication to the organization and the society, it is critical to capture employees' regular work life and phenomena.

3. Sample

I also have to say that the sample size is relatively small to support their argument on the benefits of hybrid work across industries or organizations. Suggesting the minimum number of appropriate sample sizes in scientific journals is quite controversial. However, the authors may want to justify that their sample size is sufficient to support their argument compared to other scientific articles from nature or comparable journals (e.g., Science or PNAS).

4. Practice of Hybrid Work

I am also concerned about the practice of hybrid work in the current study. The authors defined and operationalized hybrid work for working two days from home and three days at the office. However, half of the treatment group participants just had a day of WFH on mainly Friday. Whether the hybrid work practice is fully implemented or not in the current context is questionable. I assume that accounts for different findings of WFH on performance from their prior study (cf., Bloom et al.,

2015).

5. Variables

I appreciate the authors' effort in providing prototypical home office shapes and work in Figure 2 and Figure 3. However, they should consider potential confounding variables that also account for the effects of WFH. For example, the quality of the home office and surrounding factors should be considered. Not all employees are equipped with appropriate levels of work resources, such as physical home office room, computer condition, Internet speed, or information accessibility from home. Also, some employees might struggle with the presence of family members during the WFH working in the lockdown period of Shanghai 2021. The authors should consider these contextual factors.

Referee #3 (Remarks to the Author):

In the manuscript, "Hybrid Working from Home Improves Employee Retention Without Damaging Performance," the authors conduct a longitudinal field experiment to determine the impact of hybrid work on job satisfaction, quit rates, performance reviews, and managers' perceptions of hybrid work. They find that hybrid work (defined as working from home two days a week) improves job satisfaction and reduces quit rates, while not having an impact on performance grades, promotions, or lines of code written.

This manuscript makes an important and timely contribution to the literature using the methodological gold standard for research on work: a longitudinal field experiment. While the topic of study is not theoretically complex, I found the manuscript to be clearly written and the conclusions to be robust.

As the authors move forward with this manuscript, there are two key areas for improvement:

1. Discuss how the larger cultural context in which Trip.com is embedded might have influenced the results. The field experiment was conducted in Shanghai, China, which has different cultural norms and attitudes towards work (and overwork, in particular) than other countries. Could differences between countries, in terms of their cultural attitudes towards overwork, make it so that these results do not generalize to other countries? This issue needs to be discussed, and the authors need to make an argument about why these results do, or do not, generalize.
2. Communicate the results more clearly. It is tricky to communicate results when your outcome variable is a percentage, but the manuscript would be improved if the authors were clearer about when they were referring to a percentage point versus a percent.

Finally, as a small aside: I did not find extended Figure 2 or 3 to be helpful.

Author Rebuttals to Initial Comments:

Response to Referee 1

Dear Referee,

Thank you for the time you took in giving your feedback. We appreciate the time you took to read the paper, think about potential issues and explain them. This has been very helpful in improving the paper.

We have outlined above the main overall changes to the paper since the last version.

On your points in turn:

1. The authors are working on a timely and important topic. Their results have immediate policy relevance for firms around the world.

Thank you very much for the kind comment.

2. In the abstract and introduction, the authors should highlight that the effects are only statistically significant for females, non-managers and workers with commute > 90mins (highlighting the null effects reported in Figure 2)

Thank you, agreed, and we have made that change – pointed out the impact on attrition is significant for just this group – in the abstract and introduction.

3. Given that 48% employees have children, is their variation in effects for parents and non-parents?

Thank you for asking and perhaps surprisingly no. Extended Data Table 6 now runs various sample splits including the presence of children and finds no significant results on either performance grades or promotions. The likely explanation is that these employees have full-time full-week childcare. This would either be grandparents (as most employees are single children so have up to 4 grandparents to potentially provide childcare) or nurseries. So, their children would (usually) not be present at home on working from home days. As such, it is perhaps less surprising that living with young children has little impact on WFH productivity. This is also consistent with the lack of any significant difference in the volunteering rate between employees with and without children (Table 3).

4. Can the authors provide a monetary value for the 33% lower attrition (i.e., savings in costs of hiring and training?)

Thank you and yes, great idea, we have now included that in the paper in the final section. Trip.com's estimate was each quit cost about \$20,000. This aligns reasonably well with other figures in the

literature¹, that the cost of recruiting, training and developing a graduate employee is approximately 50% of their full-year salary, noting that this sample of trip.com workers had an average salary of \$68,000.

5. Clarify if the work arrangements went back to fully in-person or continued hybrid after 6 months and how this affects the interpretation of performance results?

Thank you and we have now added some text on this in the discussion section to clarify. At the end of the experiment the executives evaluated the data and decided that hybrid was positive for company profits and rolled this policy out to all employees in all divisions. As such the experiment ended in January 2022, although we do look at longer-run data in case the treatment period in 2021H2 generates longer run impacts (we did not find any longer-run differences).

6. Extended Figure 5 could show take-up rate by gender given the third policy implication mentioned in page 9?

Thank you, very interesting question, and in Figure R1.1 we now have shown the take-up rates for female and male employees – replicating Extended Figure 4 for both employees. Generally the results look similar by gender, although the take-up rate for non-volunteers is significantly higher for female than male employees (t-stat of 2.23) consistent with the suspicion that some female employees that may have wanted to WFH did not initially volunteer out of signaling concerns, although there are also other interpretations of this too (e.g. learning effects or interpretation effects of the experiment as examples). However, while the difference is statistically significant it is not that large in magnitude at 40% for men and 46% for women. We could include this figure in the extended figures if that would be helpful – we were not sure how much to push this?

7. The final sentence in the abstract ("Three days per week in the office appears to be provide sufficient in-person interaction") could be edited given that the authors do not study the effect of two days in office or one day in office?

Thank you, yes agreed, and we have deleted that sentence from the abstract. We only study 3-day hybrid WFH so cannot comment on 1-day or 2-day hybrid WFH.

¹ For example, the society for Human Resources Management, Retaining Talent, 2008, which estimates quits cost between 50% to 60% of a graduates annual salary <https://www.shrm.org/hr-today/trends-and-forecasting/special-reports-and-expert-views/documents/retaining-talent.pdf>

Figure R1.1: Take-up rates by gender

Notes: Data for 1612 employees from August 9th 2021 (volunteers) and September 13th (non-volunteers) to January 23rd 2022. Public holidays, personal holidays and excused absence (e.g. sick leave) excluded. Take-up rate is percentage of Wednesday & Friday each week they WFH.

Response to Referee 2

Dear Referee,

Thank you for the time you took in giving your feedback. We appreciate the time you took to read the paper, think about potential issues and explain them. This has been very helpful in improving the paper.

We have outlined above the main overall changes to the paper since the last version.

On your points in turn:

1) Contribution.

First of all, I would like to express my concern regarding whether and how this research really moves the needle in terms of advancing our society and organizational science (e.g., management and organizational psychology) in this area. Although I found the ideas presented to be better argued in general, I still did not find the case for the significance of the study (over and above prior work) to be sufficient. I generally agreed because the current study's findings either mirror the existing literature or could easily be intuited including their previous paper in 2015. For example, several recent papers from the management and organizational behavioral science found the benefits of hybrid work and WFH on employees' satisfaction as well as performance (e.g., Gajendran & Harrison, 2007; Gibson, Gilson, Griffith, O'Neill, 2023; Teng-Calleja, Mactal, Caringal-Go, 2023; Wood et al., 2022). Overall, I am concerned about how the current theory paper advances the literature and theory above and beyond prior research on eating and exercise.

- Gajendran, R. S., & Harrison, D. A. (2007). The good, the bad, and the unknown about telecommuting: meta-analysis of psychological mediators and individual consequences. *Journal of applied psychology*, 92(6), 1524-1541.
- Gibson, C. B., Gilson, L. L., Griffith, T. L., & O'Neill, T. A. (2023). Should employees be required to return to the office?. *Organizational Dynamics*, 52(2), 100981
- Teng-Calleja, M., Mactal, M. T. D. G., & Caringal-Go, J. F. (2023). Examining employee experiences of hybrid work: an ecological approach. *Personnel Review*.
- Wood, S., Michaelides, G., Inceoglu, I., Niven, K., Kelleher, A., Hurren, E., & Daniels, K. (2022). Satisfaction with one's job and working at home in the COVID-19 pandemic: A two-wave study. *Applied Psychology*.

We are sorry we were not clear in explaining the causal contribution of the paper in the earlier version. We have tried to improve the current version on this in the introduction highlighting that our paper provides causal evidence on the impact of hybrid, as distinct from survey or other correlational evidence.

To expand, the contribution of this paper is the first causal analysis of hybrid WFH. Hybrid is now the dominant working pattern for over 100 million employees across North America and Europe, so is an important topic in social science. This includes many academic, university, laboratory, and other research employees, many of whom are readers of *Nature*. No prior RCT or other causal analysis exists on hybrid at a scale that provides rigorous results in this setting. Indeed, because of the novelty of this paper the

2022 NBER working paper version has already received considerable academic and business interest. On that basis we would argue the research meets the high impact and novelty bar for Nature.

Thank you also for the four references you supplied to us. We have added a citation to the Wood et al (2023) paper into the discussion section of the paper to highlight the valuation of WFH experienced by employees. Having worked on the topic of working from home for 20 years and co-organized the annual Remote Work Conferences at Stanford we always appreciate references. We have summarized these four papers in Table R2.1 in relation to our paper and the literature. The key distinction vs our submission is none of these are randomized control trials, or has a causal identification strategy. They are all survey papers. There is nothing wrong with survey papers – we have written many survey papers, and indeed our monthly SWAA surveys 10,000 Americans about WFH and G-SWAA 40,000 people annually.¹ But survey papers are a different genre of paper, with a focus on conditional correlations and measurement rather than causal results.

To clarify why our Randomize Control Trial methodology is so important we quote from the earliest of the paper you cited, Ganjendran and Harrison (2007), who note on page 1533:

“It is important to note that because partial correlations do not provide conclusive evidence for the directionality of the mediated mechanism, our review of results reported below should be interpreted with caution, as representing only tentative evidence for causality.”

2) Study Context

The authors explained the study period from the second half of 2021 to the beginning of 2022 in Shanghai. As we know, Chinese policy on COVID-19 was rigorous and controlled many social functions (E.g., quarantine, lockdown, and full inspection of COVID-19). I was curious about the representativeness of the context to the normal period or endemic society. In other words, the practice of WFH and its impacts might be influenced by other contextual characteristics. Different from their previous study (Bloom, Liang, Roberts, & Ying, 2015), it is difficult to argue that the study period represents the normality. Additionally, the perception and motivation for hybrid work and WFH have changed since the COVID-19 outbreak. To make a significant scientific implication to the organization and the society, it is critical to capture employees’ regular work life and phenomena.

Thank you for highlighting your concerns here over the context, and it was our mistake to not be clearer on the conditions in Shanghai during the experiment. This arose in part because one co-author (Robin Han) was in Shanghai throughout the entire period while another (James Liang) was regularly travelling in and out of the city over this period. So, we were lulled into being too familiar with the conditions in the city during 2021. We have now included more detail in the paper for readers that may be less aware of the conditions, plus an extra exhibit in the extended figures.

¹ See www.wfhresearch.com

The summary is that yes during 2020 China did have lockdowns and again in 2022. But 2021 was a window with extremely low COVID rates so Chinese offices were operating (in Shanghai) normally. To provide evidence of this we have added some additional material.

First, Figures R2.2. and R2.3 provides photographs from the trip.com offices during the experimental period of 2021 (these are now both included in the extended figures in the paper as well). It is clear that employees are circulating freely around the offices. Some of them are wearing masks, but mostly not. So, in this sense the situation is similar to the US in spring or summer 2023, with some employees in masks but mostly not. To explain why this was the situation figure R2.4 provides the data on Covid cases rates from 2021 to early 2022. You can see case-rates in China were extremely low compared to the US because of the success of the original 2020 lockdown. They did start to rise in 2022 and that led to the second 2022 lockdown. In Figure R2.4 you can see the impact of this by looking at WFH rates of the control group. During 2021 they were coming to the office on a daily basis with complete freedom of movement. But after the COVID outbreak in early 2022 they were sent home alongside the treatment group, and the experiment ended.

We also have included a larger discussion of the context of the Chinese situation in the discussion section to provide more guidance on this. In particular, we wrote the following:

One caveat for the experiment was this was conducted in a Chinese firm based in Shanghai. While these results may not perfectly replicate in other situations, Trip.com is a large multinational with global suppliers, customers, and investors. Its offices are modern buildings that look similar to those in many American, Asian and European cities. The founding team and current senior executives have worked and lived abroad. Trip employees worked 8.6 hours per day on average, close to the 8 hours per day for graduate US employees.ⁱ As such, we believe these results that allowing employees to work from home 2-days-per week reduces quit rates with limited impacts on performance would likely extend to other organizations. “

As background for those are not familiar with the situation in China, we also wrote up the following summary we could also include in the extended in the methods section if you think that would be useful:

On 27th Jan 2020, the Shanghai government issued a return-to-work notice²: it agreed that all enterprises could resume work after 10th Feb 2020. Stricter measures were implemented for those who came back to Shanghai to work in education, childcare, medical care, housekeeping, nursing, and labor-intensive enterprises. They were put in quarantine in the dormitory of enterprises or at home for 14 days. The first day of return to work³ is 10th Feb 2020.

By 19th March, the resumption rate of supermarket outlets in Shanghai reached 99.1%; the resumption rate of shopping malls reached 100%; the resumption rate of convenience stores reached 93.4%; the

² <https://mp.weixin.qq.com/s/zXHWS5GdGuewZJIB6wRkpw>

³ <https://mp.weixin.qq.com/s/t55ktMKNWcyQ-cNcqM1bBA>

resumption rate of the catering industry reached 70%; the wholesale market of agricultural products and vegetables The market resumption rate reached 100%⁴.

Since 19 February 2021, Shanghai has turned back to a low-risk area, and all preventions and controls for COVID-19 become normalized⁵. Throughout the rest of 2021, Shanghai had been an example of “precise prevention and control”⁶, where the local authorities try best not to impose a citywide lockdown, but instead, use contact tracing to contain neighborhoods deemed high-risk. This strategy had been successful until the lockdown on 28th March 2022 due to Omicron outbreak⁷, which occurred two months after our experiment ended.

3) Sample size

I also have to say that the sample size is relatively small to support their argument on the benefits of hybrid work across industries or organizations. Suggesting the minimum number of appropriate sample sizes in scientific journals is quite controversial. However, the authors may want to justify that their sample size is sufficient to support their argument compared to other scientific articles from nature or comparable journals (e.g., Science or PNAS).

Thank for you raising this. You are correct there is no minimum sample size for publications. Instead, it is a matter of power calculations. A power calculation is performed in advance of running an RCT to confirm what size of impact you can potentially reject given your sample size and distribution of outcomes. We have attached in Figure R2.1 the power calculations that led us to select two divisions to analyze in our experiment. Trip.com has more than a dozen divisions, but our power calculations revealed a sample of about 1600 (two divisions) would be sufficient for our experiment, to reject the hypothesis of no treatment effect. The reason is our individuals have performance measures which are bounded – e.g., performance reviews are on a 1 to 5 scale while promotions are a binary variable, and are effectively independently drawn. So, a sample of 1612 is large in relation to sample standard deviations. Thus, our confidence intervals around our null effects are small, and the experiment has sufficient power to reject any material performance impact.

More generally we disagree with the statement “*the sample size is relatively small*”. This sample size is larger than 3 of the papers you cited. The fourth paper (Gajendran and Harrison, 2007) pools across studies, many of them unpublished, so is not comparable. Our paper’s sample size is also larger than 602 lab experiment sample and similar to the 1490 engineer sample in the Brucks and Levav (2022) paper published by Nature which is perhaps closest to this paper. It is also substantially larger than the most cited WFH RCT paper by Bloom, Liang, Roberts and Ying (2015), who study 249 employees.

⁴ https://www.shanghai.gov.cn/nw48546/20200826/0001-48546_1434204.html

⁵ See government website, archived: <https://archive.vn/9Vlze>.

⁶ <https://www.globaltimes.cn/page/202101/1213974.shtml>

⁷ http://www.legaldaily.com.cn/commentary/content/2022-01/20/content_8661029.htm

More generally, we downloaded the entire American Economic Association RCT registry⁸ as a large easy to analyze sample of experiments which on 10/31/2023 contained 7,254 Randomized Control Trials. The median sample size in this was 1197, slightly smaller than the sample size of our experiment.

So, on that basis I think our sample size is both sufficient to investigate the key hypothesis in the paper, and in absolute terms is moderately large. Thank you for letting us clarify this.

4) Practice of Hybrid Work

I am also concerned about the practice of hybrid work in the current study. The authors defined and operationalized hybrid work for working two days from home and three days at the office. However, half of the treatment group participants just had a day of WFH on mainly Friday. Whether the hybrid work practice is fully implemented or not in the current context is questionable. I assume that accounts for different findings of WFH on performance from their prior study (cf., Bloom et al., 2015).

Thank you for highlighting concerns that the study did not force employees to WFH on Wednesday and Friday, but allowed them to choose, resulting in some employees coming into work on eligible WFH days.

We see this as an attribute rather than an issue with the study. In practice, very few US and European employers force employees to WFH on their home days. This would be extremely problematic for employees that lack suitable working conditions at home, or may feel isolated working at home. For example, employees living alone, or with poor home internet, noisy working conditions, or sharing living space with young children or flat mates may not be happy (or productive) being forced to work from home on their home days. Having talked to 100s of organizations about hybrid WFH the standard practice is to give employees the option but not the obligation to WFH on their home days. Most employees take up that option – similar to the employees in our study – but not all employees.

So, thank you for noting this. We see it as a strength of our study that it tests a realistic hybrid WFH policy that is most common in North American and European organizations.

5) Variables

I appreciate the authors' effort in providing prototypical home office shapes and work in Figure 2 and Figure 3. However, they should consider potential confounding variables that also account for the effects of WFH. For example, the quality of the home office and surrounding factors should be considered. Not all employees are equipped with appropriate levels of work resources, such as physical home office room, computer condition, Internet speed, or information accessibility from home. Also, some employees might struggle with the presence of family members during the WFH working in the lockdown period of Shanghai 2021. The authors should consider these contextual factors.

Thanks you and yes, we completely agree variables are an important factor shaping the potential impact of WFH. Indeed, in reflection of this we collected data in surveys and from company administrative data on the issues you highlighted. It was our mistake not to be clearer on this in the original submission. We

⁸ <https://www.aeaweb.org/journals/policies/rct-registry>

have data on children, commuting time, living alone, children at home and internet speed. Since these interactions were all insignificant these results were either in the appendix in the prior version or not included. However, we have now created a larger omnibus table with these interactions in Table R2.6 and have included these now in the extended tests in the updated paper. As you see these are potentially important co-founders but there are no notably robust interactions with them and the promotions or performance impact of hybrid WFH.

ⁱ US Bureau of Labor Statistics American Time Use Survey <https://www.bls.gov/charts/american-time-use/emp-by-ftp-job-edu-h.htm>

Table R2.1: Summary of the Four Papers Mentioned by Referee 2

Paper	Randomized Control Trial	Sample size	Summary	Published in a Journal	Year
Gajendran, R. S., & Harrison, D. A. (2007). The good, the bad, and the unknown about telecommuting: meta-analysis of psychological mediators and individual consequences. Journal of applied psychology, 92(6), 1524-1541.	No	46 studies, spans 12,883 underlying observations	Literature review of surveys and interviews. For example, notes on page 1531 that: "Each of the 46 studies in our final sample used survey or interview techniques to measure the proximal and distal outcomes of telecommuting". Also notes on page 1536 "There were no randomized, highly controlled field experiments in any of our collected studies, and, indeed, it would be impossible to create placebo or double-blind conditions for such an elaborate work arrangement as telecommuting."	Overview published in Journal of Applied Psychology . 27 of the underlying were journal published and 19 were unpublished	2007
Wood, S., Michaelides, G., Inceoglu, I., Niven, K., Kelleher, A., Hurren, E., & Daniels, K. (2022). Satisfaction with one's job and working at home in the COVID-19 pandemic: A two-wave study. Applied Psychology.	No	1173	Survey of academics from two English universities on relationship between WFH and various factors like job-satisfaction.	Yes - Applied Psychology	2022
Teng-Calleja, M., Mactal, M. T. D. G., & Caringal-Go, J. F. (2023). Examining employee experiences of hybrid work: an ecological approach. Personnel Review.	No	45	Survey of 45 Filipino employees on their experiences working in a hybrid environment	Yes - Emerald Insights	2023
Gibson, C. B., Gilson, L. L., Griffith, T. L., & O'Neill, T. A. (2023). Should employees be required to return to the office?. Organizational Dynamics, 52(2), 100981	No	4 interviews	Interview of 4 academics in US and Canadian universities on employees returning to the office	Yes - Organizational Dynamics	2023

Figure R2.1 – Power Calculations

For any test the p-value is the inverse t-statistic of the (difference/standard-error). Given iid distribution in the treatment and control – randomization is undertaken at the individual level – the standard-error= $n^{(-1/2)} * \sigma$ where n would be the treatment and control sample sizes and sigma individual performance sampling error.

So, what difference in performance grade could we reject with a two division (1600 person) sample? The performance grade has a 5 point grading scale, with a mean of approximately 4 and a standard deviation of approximately 1. Then the standard error of the treatment and control group sample means, which are both around 800 observations given a randomized split of our overall sample of approximately 1600, will be approximately $800^{(-1/2)} * 1 = .03$. The standard-error of the difference then adds the standard errors so this would be 0.06.

A difference/standard error ratio of 1.96 of this would typically be expected to be statistically significant, which implies a difference between treatment and control of approximately 0.15. This is the level of performance difference in promotion scores we can expect to identify if it existed between treatment and control. In words, we could expect to reject the null of equality for treatment differences of about 0.15 or greater, which on a 1 to 5 performance scale would be the same as, for example, 20 employees in the treatment sample getting a score of 4 and 17 employees in the control sample getting a score of 4 and 3 of getting a 5.

To complicate matters we actually have multiple different outcomes – performance grades, promotions, lines of code written and various interaction effects. Hence, if we are testing all of these and accepting any rejection of equality as a treatment effect then the power of our test is substantially greater. It is hard to know how much greater because of correlations across tests, but certainly we can reject any substantial differences in performance – say if 20% of employees get a one grade higher performance review.

On this basis we conclude two divisions at around 1600 employees is a sufficiently large sample to detect any material impact of the WFH hybrid treatment on employee performance. Larger samples adding additional divisions would not be necessary given the additional complexities and experimental challenges this would involve.

Figure R2.2: The company had modern open-plan offices, with teams seated together

Figure R2.3: The company had modern offices with teams working and training together mostly in person

Figures from top-left clockwise:
1) Team meeting describing the experiment, August 2021
2) An employee training camp, September 2021
3) An employee workshop, October 2021

Source: trip.com internal photographs

Figure R2.4 In 2021 China had a low covid case rate, so there was no required office masking or distancing

Daily Covid-19 cases per million people

Notes: From the WHO COVID-19 Dashboard <https://covid19.who.int/info>

Figure R2.5 The control group came into work every day until the January 2022 Covid outbreak

Notes: Data for 1612 employees from August 9th 2021 (volunteers) and September 13th (non-volunteers) to January 23rd 2022. Public holidays, personal holidays and excused absence (e.g. sick leave) excluded. Take-up rate is percentage of Wednesday & Friday each week they WFH.

Figure R2.6 Extended Data with Context Interactions - no robust significant treatment heterogeneity

	(1)	(2)	(3)	(4)	(5)	(6)	(7)	(8)	(9)	(10)	(11)
Treat	-0.056	0.011	-0.070	-0.032	-0.075	-0.066	-0.030	-0.047	-0.064	-0.057	-0.087
	{0.198}	{0.860}	{0.302}	{0.602}	{0.158}	{0.177}	{0.614}	{0.433}	{0.223}	{0.517}	{0.170}
Children		-0.043									
		{0.477}									
Treat*Children		-0.151									
		{0.079}									
Long commute			-0.129								
			{0.061}								
Treat*Long commute			0.029								
			{0.768}								
Engineer				-0.078							
				{0.200}							
Treat*Engineer				-0.051							
				{0.554}							
Female					-0.127						
					{0.048}						
Treat*Female					0.064						
					{0.483}						
Manager						0.286					
						{0.000}					
Treat*Manager						0.058					
						{0.552}					
Long tenure							-0.133				
							{0.028}				
Treat*Long tenure							-0.065				
							{0.448}				
Treat manager								0.043			
								{0.483}			
Treat*Treat manager								-0.021			
								{0.810}			
Volunteer									-0.081		
									{0.219}		
Treat*Volunteer									0.029		
									{0.757}		
Living Alone										-0.121	
										{0.380}	
Treat*Living Alone										0.093	
										{0.636}	
Super Internet											0.001
											{0.987}
Treat*Super Internet											0.048
											{0.613}
N	1507	1507	1189	1507	1507	1507	1507	1507	1507	481	1244

	(1)	(2)	(3)	(4)	(5)	(6)	(7)	(8)	(9)	(10)	(11)
Treat	0.009	-0.007	-0.004	-0.003	0.020	-0.000	0.014	0.024	0.011	0.008	0.001
	{0.503}	{0.723}	{0.838}	{0.880}	{0.224}	{0.992}	{0.460}	{0.193}	{0.518}	{0.748}	{0.942}
Children		-0.081									
		{0.000}									
Treat*Children		0.023									
		{0.389}									
Long commute			-0.052								
			{0.018}								
Treat*Long commute			0.028								
			{0.367}								
Engineer				0.051							
				{0.008}							
Treat*Engineer				0.025							
				{0.344}							
Female					0.007						
					{0.711}						
Treat*Female					-0.032						
					{0.261}						
Manager						-0.067					
						{0.002}					
Treat*Manager						0.034					
						{0.274}					
Long tenure							-0.058				
							{0.002}				
Treat*Long tenure							-0.015				
							{0.568}				
Treat manager								0.036			
								{0.063}			
Treat*Treat manager								-0.034			
								{0.213}			
Volunteer									-0.003		
									{0.903}		
Treat*Volunteer									-0.005		
									{0.875}		
Living Alone										0.121	
										{0.003}	
Treat*Living Alone										-0.023	
										{0.693}	
Super Internet											-0.033
											{0.141}
Treat*Super Internet											-0.001
											{0.980}
N	1507	1507	1189	1507	1507	1507	1507	1507	1507	481	1244

Notes: Regression result tables, with each column being one regression. The dependent variables are the performance measure after assigning each letter grade a numeric value from 1 (D) to 5 (A) (left) and promotion indicator (right). Children is an indicator for having children and long commute is an indicator for commuters with above median commute time, which is missing in unreported cases. All others are binary definitions. P-values for two-tailed t- tests are reported in curly brackets. Sample sizes in the row “N” at the foot of the table may be lower than the full-sample if data is only present for sub-samples (e.g survey respondents). No other covariates are included.

Response to Referee 3

Dear Referee,

Thank you for the time you took in giving your feedback. We appreciate the time you took to read the paper, think about potential issues and explain them. This has been very helpful in improving the paper.

On the points in turn:

This manuscript makes an important and timely contribution to the literature using the methodological gold standard for research on work: a longitudinal field experiment. While the topic of study is not theoretically complex, I found the manuscript to be clearly written and the conclusions to be robust.

Many thanks for the very kind and supportive comments.

1. Discuss how the larger cultural context in which Trip.com is embedded might have influenced the results. The field experiment was conducted in Shanghai, China, which has different cultural norms and attitudes towards work (and overwork, in particular) than other countries. Could differences between countries, in terms of their cultural attitudes towards overwork, make it so that these results do not generalize to other countries? This issue needs to be discussed, and the authors need to make an argument about why these results do, or do not, generalize.

Thank you and agreed. We now add a paragraph to note this difference in the discussion section. In particular, we write the following.

One caveat for the experiment was this was conducted in a Chinese firm based in Shanghai. While these results may not perfectly replicate in other situations, Trip.com is a large multinational with global suppliers, customers, and investors. Its offices are modern buildings that look similar to those in many American, Asian and European cities. The founding team and current senior executives have worked and lived abroad. Trip employees work 8.6 hours per day on average, close to the 8 hours per day for graduate US employees.¹ As such, we believe these results that allowing employees to work from home 2-days-per week reduces quit rates with limited impacts on performance would likely extend to other organizations.

¹ Trip.com based on calculations using office swipe-card data for work on-site plus VPN and survey data to add hours worked from home. US data from the American Time Use Survey <https://www.bls.gov/charts/american-time-use/emp-by-ftpt-job-edu-h.htm>

Also to note in terms of working hours there is a weekly working hour ranking of tech firm in China, where individual employees can self-report their workload (so this number possibly overestimates the average although hopefully the ranking is informative)². According to the ranking, Trip has an average 49.2 hours per week, ranking 39th in China on 1st June 2023; 46.5 hours per week, ranked 44th as of 20th August, 2022³, and 45.8 hours per week, ranking 38th on 23rd Nov 2021⁴.

2. Communicate the results more clearly. It is tricky to communicate results when your outcome variable is a percentage, but the manuscript would be improved if the authors were clearer about when they were referring to a percentage point versus a percent.

Thanks, totally agree and great point. Rereading this now this is indeed confusing, and we have tried to either use words like “a third” which we think is clearer, or if not stick to percentage impacts and then in brackets simply statement treatment and control outcomes. Much appreciated for pointing this out – clarity is important given the hopefully broader readership for this article.

Finally, as a small aside: I did not find extended Figure 2 or 3 to be helpful.

Thank you for the feedback and we have taken out Extended Figure 2. The reason for retaining extended figure 3 was to highlight the conditions in employees’ homes, which typically were not large home-offices (as may be the case for senior managers in the US, or indeed many tenured faculty in US universities) but instead locations on kitchen tables within modest apartments. We would be happy to take this figure out too if you thought that would be helpful, but we just wanted to give some context for why this remained in the updated paper.

² <https://duibiao.info/post/4211>

³ <https://www.iyunying.org/news/289682.html>

⁴ <https://t.cj.sina.com.cn/articles/view/2708017977/a169133900100x7wc>

Reviewer Reports on the First Revision:

Referees' comments:

Referee #1:

Remarks to the Author:

I have no other outstanding comments. Thanks for addressing my prior comments and suggestions.

Referee #2:

Remarks to the Author:

Thank you for addressing all my comments. I also have enjoyed and learned a lot from the current paper. First, thank you for providing me with an explanation of the context.

I have only one comment. I was continuously curious about the external validity of the current study and whether we can apply the findings to normal or other situations. Trip.com is a global online travel agency that makes profits from the travel business. As we know, 2021 might be difficult for all travel-related industries, and there might be less work. I am curious whether the hybrid work and its effects are the same during the normal periods with more work demands. Now, at the beginning of 2024, there might be more work requiring more interactions or workplace dynamics. Please elaborate on how the current study and findings represent the other contexts.

Referee #3:

Remarks to the Author:

This research makes an important and timely contribution to the literature, and the authors have done an excellent job of addressing the concerns raised by myself and the other reviewers.